# Causal Preference Elicitation

Edwin V. Bonilla [1]   He Zhao [1]   Daniel M. Steinberg [1]

## Abstract

We propose causal preference elicitation, a Bayesian framework for expert-in-the-loop causal discovery that actively queries local edge relations to concentrate a posterior over directed acyclic graphs (DAGs). From any black-box observational posterior, we model noisy expert judgments with a three-way likelihood over edge existence and direction. Posterior inference uses a flexible particle approximation, and queries are selected by an efficient expected information gain criterion on the expert's categorical response. Experiments on synthetic graphs, protein signaling data, and a human gene perturbation benchmark show faster posterior concentration and improved recovery of directed effects under tight query budgets.

## 1. Introduction

Causal structure learning from observational data underpins reasoning and decision making in many areas of science, medicine, and public policy (Spirtes et al., 2001; Pearl, 2009; Peters et al., 2017). A causal graph makes explicit which variables directly influence others, thereby supporting counterfactual reasoning and policy evaluation rather than mere prediction (Pearl, 2009). However, inferring such graphs from finite, noisy observational data is notoriously difficult: many distinct directed acyclic graphs (DAGs) are Markov equivalent, likelihood surfaces are highly multi-modal, and traditional discovery methods often remain agnostic about large portions of the structure (Spirtes et al., 2001; Chickering, 2002). Therefore, in practice, causal discovery is rarely a purely data-driven exercise; practitioners routinely consult domain experts who possess qualitative knowledge about plausible mechanisms, forbidden edges, or expected directions of influence (Heckerman et al., 1995; Meek, 1995).

Existing approaches for incorporating expert knowledge into causal discovery are typically *static* and *coarse*. Prior

information is encoded as hard constraints, simple edge-wise priors, or post-hoc regularizers that bias learning toward expert-specified graphs (Cooper & Herskovits, 1992; Heckerman et al., 1995; Kuipers et al., 2022; Zheng et al., 2018). Such treatments usually assume expert knowledge is uniformly reliable, ignoring uncertainty or systematic bias, and they treat expert input as a one-shot specification, without asking which *additional* judgments would be most informative despite expert time being scarce. This contrasts with work on active structure learning and causal interventions, which optimizes experimental choices but typically does not model subjective expert judgments or their uncertainty (Tong & Koller, 2001; Eberhardt, 2008; Hauser & Bühlmann, 2012).

In this work we introduce CaPE, a probabilistic, sequential framework for *causal preference elicitation*: actively querying an expert about local edge relations to concentrate a posterior distribution over directed acyclic graphs (DAGs). We assume access to samples from an initial observational posterior $q_0(W \mid X)$, obtained from any black-box causal discovery method, including Bayesian structure learning, score-based MCMC, bootstrap aggregation, or variational DAG approaches (Madigan & York, 1995; Friedman & Koller, 2003; Kuipers et al., 2022; Zheng et al., 2018; Thompson et al., 2025). Expert knowledge is modeled via a categorical response matrix $Y$ encoding (possibly noisy) judgments about edge existence and orientation; the learner sequentially selects pairs of variables $(i, j)$ and observes the corresponding responses $Y_{ij}$. The central challenge is to choose queries that maximally reduce posterior uncertainty over the *entire* graph, in the sense of information-efficient Bayesian active learning (MacKay, 1992; Sebastiani & Wynn, 2000).

To this end, we introduce an explicit probabilistic expert model $p_\theta(Y \mid W)$ that maps weighted DAGs to three-way categorical responses using interpretable direction scores based on edge weights and optional structural features, with logistic components controlling reliability and decisiveness. This allows expert feedback to be informative, heterogeneous, and uncertain; consistent with probabilistic models of noisy annotators and preference judgments (Houlsby et al., 2011; Kim & Ghahramani, 2012). Posterior inference is performed using a particle-based approximation updated via importance reweighting, effective-sample-size-based resampling, and lightweight MCMC rejuvenation to preserve

[1]CSIRO, Australia. Correspondence to: Edwin V. Bonilla <edwin.bonilla@csiro.au>.

*Proceedings of the 43$^{rd}$ International Conference on Machine Learning*, Seoul, South Korea. PMLR 306, 2026. Copyright 2026 by the author(s).

diversity while enforcing acyclicity (Doucet et al., 2001; Del Moral et al., 2006; Madigan & York, 1995). Finally, our query selection strategy is driven by a Bayesian active learning by disagreement (BALD, Houlsby et al., 2011) formulation of the expected information gain (EIG) criterion. This is computed from posterior predictive label distributions, enabling efficient selection of locally queried edges with global structural impact (Houlsby et al., 2011; Tong & Koller, 2001; Sebastiani & Wynn, 2000).

CaPE is complementary to Bayesian optimal experimental design for causal discovery (Lindley, 1956; Chaloner & Verdinelli, 1995; Tigas et al., 2023; Annadani et al., 2024). Intervention-design methods choose experimental actions and observe new interventional data, whereas CaPE chooses structural questions and observes noisy categorical judgments through an expert channel. This distinction is important in domains where interventions are expensive, ethically constrained, or already fixed, but domain experts can still provide local mechanistic knowledge.

Our main **contributions** can be summarized as follows:

- We propose a general framework for expert-in-the-loop causal discovery cast as a problem of causal preference elicitation, combining an arbitrary observational posterior $q_0(W \mid X)$ with sequential expert feedback about local edge relations, connecting and extending prior work on expert-augmented causal discovery and active structure learning (Heckerman et al., 1995; Tong & Koller, 2001).

- We introduce a flexible hierarchical logistic expert model that assigns three-way categorical labels (edge $i \rightarrow j$, edge $j \rightarrow i$, or no edge) based on local direction scores. While these scores are defined in terms of edge weights in our experiments, the model is general and can incorporate graph-structural features such as v-structure support or cycle risk, following feature-based approaches to edge orientation (Meek, 1995; Tsamardinos et al., 2006).

- We develop a particle-based Bayesian updating scheme for DAG posteriors that systematically integrates expert responses via importance reweighting, ESS-triggered resampling, and Metropolis-Hastings rejuvenation while strictly enforcing acyclicity, leveraging sequential Monte Carlo ideas in a new expert-in-the-loop setting (Doucet et al., 2001; Del Moral et al., 2006).

- We derive a BALD-style expected information gain objective for query selection that operates on low-dimensional categorical distributions, making information-efficient active learning on DAG space computationally tractable (Houlsby et al., 2011; MacKay, 1992).

- We provide experiments demonstrating that our EIG-based acquisition policy substantially accelerates posterior concentration towards the true graph compared to

random and uncertainty-based baselines, improving both probabilistic predictive scores and structural metrics. Additional misspecification studies show that the method remains competitive under several realistic non-adversarial deviations from the assumed expert model.

Taken together, these components provide a principled approach for transforming qualitative expert intuitions about "what causes what" into a carefully budgeted sequence of targeted questions, closing the loop between probabilistic causal inference and expert human judgment.

Code and reproducibility scripts are publicly available at https://github.com/csiro-funml/CaPE.

## 2. Related Work

Causal discovery methods typically infer DAGs from observational data using constraint-based or score-based approaches (Pearl, 2009; Spirtes et al., 2001). Recent work has proposed continuous characterizations of acyclicity to enable scalable optimization (Zheng et al., 2018; Bello et al., 2022; Andrews et al., 2023), while Bayesian formulations maintain posterior distributions over structures and provide principled uncertainty quantification (Thompson et al., 2025; Annadani et al., 2023; Bonilla et al., 2026).

Incorporating domain expertise into Bayesian network and causal structure learning has a long history, most commonly through structural constraints or priors over graphs and parameters (Heckerman et al., 1995; Cooper & Herskovits, 1992; de Campos et al., 2009; Meek, 1995). More expressive forms of expert knowledge (including path, ancestral, grouping, ordering, and type constraints) have been shown to substantially reduce the space of admissible DAGs and improve recovery accuracy (Borboudakis & Tsamardinos, 2012; Chen et al., 2016; Parviainen & Kaski, 2017; Shojaie & Chen, 2024; Brouillard et al., 2022). Despite their practical importance, such knowledge is typically specified *a priori* and treated as fixed during learning (Kitson et al., 2023).

A parallel literature studies Bayesian experimental design and active causal discovery, where the learner selects interventions, experiments, or queries to reduce uncertainty over causal structures (Lindley, 1956; Chaloner & Verdinelli, 1995; Tong & Koller, 2001; Murphy, 2001; Toth et al., 2022; Tigas et al., 2023; Annadani et al., 2024). These methods usually acquire new data through interventions, while CaPE acquires local structural information through noisy expert responses. The two settings share information-gain principles but differ in action space, observation model, and deployment regime. Recent work highlights the role of expert advice and iterative human interaction in refining causal models (Choo et al., 2023; Gururaghavendran & Murray, 2024; Kitson & Constantinou, 2025; Ankan & Textor, 2025;

Ou et al., 2024; da Silva et al., 2025). Relative to these approaches, CaPE is not restricted to linear structural equation models (SEMs) and emphasizes modular posterior refinement: it requires only posterior samples over DAGs, uses an explicit three-way probabilistic expert likelihood for local edge relations, and selects queries by a tractable BALD-style mutual-information objective. These approaches and additional related work are further discussed in Section F.

## 3. Setup

Let $X \in \mathbb{R}^{N \times D}$ denote an observational dataset consisting of $N$ i.i.d. samples from an unknown causal data-generating process. A causal graph on $D$ nodes is represented by a matrix $W \in \mathbb{R}^{D \times D}$, $W_{ii} = 0$ for all $i \in [D]$, whose entries encode direct causal effects ($[D] := \{1, \ldots, D\}$). The binary adjacency matrix $A = \mathbf{1}(W \neq 0) \in \{0,1\}^{D \times D}$ is required to be acyclic, so $W$ represents a weighted directed acyclic graph (DAG). We do not assume a specific structural equation model linking $W$ to the observational data $X$; only that $X$ yields information about the underlying DAG.

**Initial posterior.** From the observational dataset we assume access to samples from an initial posterior distribution $q_0(W \mid X)$, obtained by any standard causal discovery method (e.g., MCMC, bootstrap aggregation, or variational learning). This distribution captures our uncertainty about the causal structure before consulting an expert.

**Expert-response random matrix.** We introduce a random matrix $Y \in \{0,1,2\}^{D \times D}$, $Y_{ii} = 2$ for all $i \in [D]$, representing the (possibly noisy) judgments of an expert about each ordered pair $(i,j)$. Each entry takes values in the alphabet $\mathcal{Y} = \{0,1,2\}$, with the semantic interpretation $Y_{ij} = 1$ (direct edge $i \to j$), $Y_{ij} = 0$ (direct edge $j \to i$), $Y_{ij} = 2$ (no direct edge). The conditional distribution of $Y$ given $W$ is governed by an expert-likelihood model $p_\theta(Y \mid W)$, specified later in Section 4. We do not assume the expert is deterministic or correct.

**Sequential revelation of expert judgments.** The learner does not observe the full matrix $Y$. Instead, at each round $t = 1, 2, \ldots$, the learner selects an ordered pair $(i_t, j_t)$, $i_t \neq j_t$, and is given the corresponding entry of $Y$, i.e., $Y_{i_t j_t}^{(t)} = Y_{i_t j_t}$. Thus $Y$ is revealed one coordinate at a time. The information available up to round $t$ is the set

$$\mathcal{D}_{1:t} = \big\{(i_\tau, j_\tau, Y_{i_\tau j_\tau}^{(\tau)}) : \tau = 1, \ldots, t\big\} \subseteq [D]^2 \times \mathcal{Y},$$

representing a progressively revealed subset of entries of the latent expert-response matrix.

**Goal.** Given the observational data $X$, samples from an initial posterior $q_0(W \mid X)$, and sequential access to expert judgments through the observed entries $\mathcal{D}_{1:t}$, our objective is to recover the underlying DAG under the assumed causal

model[1] or to concentrate the posterior distribution over $W$ around its true value. The central design question is how to choose the next pair $(i_t, j_t)$ whose entry $Y_{ij}$ will be revealed. This is formulated as an active learning problem on DAG space, where the learner selects queries to maximally reduce posterior uncertainty.

## 4. Expert Likelihood Model

We now specify the probabilistic model $p_\theta(Y_{ij} \mid W)$ describing how the expert generates judgments about each ordered pair $(i,j)$ from an underlying weighted DAG $W$. The model maps the latent graph to the categorical edge label $Y_{ij} \in \{0,1,2\}$ corresponding to $\{j \to i, \ i \to j, \ \varnothing\}$. The expert may be uncertain or incorrect; the likelihood need not be deterministic.

### 4.1. Local Direction Scores

For any candidate graph $W$ and any ordered pair $(i,j)$ with $i \neq j$, we define a *local direction score*

$$s_{i \to j}(W) = g\left(\frac{|W_{ij}|}{\gamma}\right) + \lambda \phi_{i \to j}(W), \qquad (1)$$

where: $g : (0, \infty) \to \mathbb{R}$ is a monotone link function capturing evidence from the edge magnitude. In all experiments we use $g(u) = \log(\varepsilon + u), \varepsilon > 0$, which yields more negative scores for very small weights; $\gamma > 0$ is a scale parameter determining how large an edge weight must be to constitute meaningful evidence; $\phi_{i \to j}(W)$ is an optional structural feature reflecting properties of the graph $W$ such as v-structure support, cycle risk, or posterior orientation odds (see Appendix A); and $\lambda \geq 0$ controls the contribution of the structural feature.

The two scores $s_{i \to j}(W)$ and $s_{j \to i}(W)$ are combined to produce two summary statistics:

$$a_{ij}(W) = \max\{s_{i \to j}(W), \ s_{j \to i}(W)\}, \quad (2)$$
$$d_{ij}(W) = s_{i \to j}(W) - s_{j \to i}(W). \quad (3)$$

The quantity $a_{ij}(W)$ expresses the absolute evidence that *some* direct edge exists between $i$ and $j$, while $d_{ij}(W)$ measures the relative evidence in favor of one orientation over the other.

### 4.2. Hierarchical Logistic Expert Model

Let $\theta = (\beta_{\mathrm{edge}}, \beta_{\mathrm{dir}}, \lambda, \gamma)$ collect the expert hyperparameters. For each ordered pair $(i,j)$ and candidate graph $W$ we

---

[1]Throughout, we interpret the DAG as encoding causal assumptions (e.g., causal sufficiency and the Markov and faithfulness conditions), rather than as an object whose causal meaning is identified purely from observational data. Our goal is, therefore, to concentrate posterior mass on the true graph under these assumptions, not to claim causal discovery in the absence of them.

introduce two logistic components:

**Edge-existence probability.** The probability that the expert believes there is a direct edge between $i$ and $j$ is

$$p_{\text{edge}}(W, i, j) = \sigma\big(\beta_{\text{edge}}\, a_{ij}(W)\big), \qquad (4)$$

where $\sigma(u) = 1/(1 + e^{-u})$ is the logistic sigmoid and $\beta_{\text{edge}} > 0$ determines the sharpness of the expert's ability to distinguish "edge" from "no edge."

**Direction probability conditional on existence.** Given that a direct edge is believed to exist, the probability that the expert orients it as $i \rightarrow j$ is

$$p_{i \rightarrow j | \text{edge}}(W, i, j) = \sigma\big(\beta_{\text{dir}}\, d_{ij}(W)\big), \qquad (5)$$

where $\beta_{\text{dir}} > 0$ governs how sharply the expert distinguishes between the two possible orientations. The reverse orientation has probability

$$p_{j \rightarrow i | \text{edge}}(W, i, j) = 1 - p_{i \rightarrow j | \text{edge}}(W, i, j). \qquad (6)$$

### 4.3. Three-way Likelihood for $Y_{ij}$

Combining the two logistic components yields a three-class categorical distribution over $\{0, 1, 2\}$:

$$p_\theta(Y_{ij} = 2 \mid W) = 1 - p_{\text{edge}}(W, i, j), \qquad (7)$$

$$p_\theta(Y_{ij} = 1 \mid W) = p_{\text{edge}}(W, i, j)\, p_{i \rightarrow j | \text{edge}}(W, i, j), \quad (8)$$

$$p_\theta(Y_{ij} = 0 \mid W) = p_{\text{edge}}(W, i, j)\, p_{j \rightarrow i | \text{edge}}(W, i, j). \quad (9)$$

Thus, intuitively, this hierarchical likelihood expert-model first decides whether a direct edge exists, and if so, chooses its orientation. The parameters $(\beta_{\text{edge}}, \beta_{\text{dir}})$ determine the reliability and decisiveness of the expert.

## 5. Posterior Representation and Update

We now describe how expert feedback is incorporated into a posterior distribution over weighted DAGs. Recall that at each round $t$ the learner observes a single entry $Y_{i_t j_t}^{(t)}$ of the latent expert-response matrix $Y$. As before, let $\mathcal{D}_{1:t} = \{(i_\tau, j_\tau, Y_{i_\tau j_\tau}^{(\tau)}) : \tau = 1, \ldots, t\}$ denote all revealed entries up to time $t$. Given the observational posterior $q_0(W \mid X)$ and the expert-likelihood model $p_\theta(Y_{ij} \mid W)$ introduced in Section 4, the goal is to maintain the evolving posterior distribution $q_t(W) := q_t(W \mid X, \mathcal{D}_{1:t})$.

### 5.1. Bayesian Update

Because the expert responses are conditionally independent given $W$, the posterior admits the factorized form

$$q_t(W) \propto q_0(W \mid X) \prod_{\tau=1}^{t} p_\theta\left(Y_{i_\tau j_\tau}^{(\tau)} \mid W\right). \qquad (10)$$

Equivalently, the update from $t - 1$ to $t$ is

$$q_t(W) \propto q_{t-1}(W)\, p_\theta\left(Y_{i_t j_t}^{(t)} \mid W\right).$$

This update is fully Bayesian and requires only the likelihood of the newly revealed entry of $Y$. The primary challenge is that the space of DAGs grows superexponentially in $D$, making exact posterior computation intractable. We therefore adopt a particle-based approximation.

### 5.2. Particle Representation

We represent the posterior at round $t$ by a weighted empirical distribution

$$q_t(W) \approx \sum_{s=1}^{S} w_t^{(s)} \delta_{W^{(s)}}(W), \qquad (11)$$

where $\{W^{(s)}\}_{s=1}^{S}$ are particles (candidate weighted DAGs) and $\{w_t^{(s)}\}_{s=1}^{S}$ are their corresponding normalized importance weights. At initialization, $W^{(s)} \sim q_0(W \mid X), w_0^{(s)} = \frac{1}{S}$, where we note that we only require samples from the initial posterior rather than access to the actual density in closed form. This particle representation is useful for causal discovery because posterior uncertainty over DAGs is often multimodal, as each expert response induces a simple sequential Bayesian weight update, and because the approximation is agnostic to how the initial particles were obtained. Although we use data-driven posteriors in the real-data experiments, the same machinery can start from any prior distribution over DAGs, including weak or non-informative priors; observational initialization mainly improves query efficiency by focusing elicitation on ambiguities left by the data.

**Weight update.** Upon observing $Y_{i_t j_t}^{(t)}$, the weights are updated by

$$w_t^{(s)} \propto w_{t-1}^{(s)} p_\theta\left(Y_{i_t j_t}^{(t)} \mid W^{(s)}\right), \qquad \sum_{s=1}^{S} w_t^{(s)} = 1.$$

This update increases the weight of particles whose local structure agrees with the expert feedback and decreases the weight of those that disagree.

### 5.3. Effective Sample Size and Resampling

After several rounds, the particle weights may become concentrated on a small subset of particles. We monitor particle degeneracy using the effective sample size (ESS) $\text{ESS}(w_t) = \frac{1}{\sum_{s=1}^{S}(w_t^{(s)})^2}$. When $\text{ESS}(w_t) < \delta_s S$ for a threshold $\delta_s \in (0, 1)$, we resample the particles according to their weights, producing a uniformly-weighted set $\{W^{(s)}\}_{s=1}^{S} \sim q_t(W)$ with $w_t^{(s)} = 1/S$.

### 5.4. Particle Rejuvenation

Resampling reduces weight degeneracy but may create duplicate particles and diminish diversity. To mitigate this, we apply a lightweight Metropolis–Hastings rejuvenation kernel $K_t(W' \mid W)$ to each resampled particle.

A proposal $W'$ is generated from $W$ by a small random graph edit, such as: adding or removing a single directed edge; flipping an existing edge orientation; or adjusting the weight of a random edge. Proposals violating acyclicity are rejected immediately. Otherwise the proposal is accepted with probability

$$\alpha(W', W) = \min\left\{1, \; \frac{q_0(W' \mid X) \prod_{\tau=1}^{t} p_\theta(Y_{i_\tau j_\tau}^{(\tau)} \mid W')}{q_0(W \mid X) \prod_{\tau=1}^{t} p_\theta(Y_{i_\tau j_\tau}^{(\tau)} \mid W)}\right\}.$$

This leaves $q_t$ invariant and restores diversity among particles as the posterior concentrates.

**Summary.** The combination of an importance-weight update, ESS-triggered resampling, and rejuvenation yields a flexible particle approximation to the evolving posterior over DAGs, enabling active learning and iterative expert refinement in high-dimensional graph spaces.

## 6. Query Selection and Acquisition Policies

At each round the learner must choose an ordered pair $(i, j)$ corresponding to variables $x_i, x_j$, $i \neq j$, whose expert-response entry $Y_{ij}$ will be revealed. The choice of query should maximally reduce uncertainty about the latent DAG $W$. Below we describe how to achieve this.

### 6.1. Posterior Predictive Distribution

For any fixed ordered pair $(i, j)$ and any label $y \in \{0, 1, 2\}$, the posterior predictive probability of observing $Y_{ij} = y$ at round $t + 1$ is

$$\widehat{p}_t^{ij}(y) := \mathbb{E}_{W \sim q_t}[p_\theta(Y_{ij} = y \mid W)]$$
$$= \sum_{s=1}^{S} w_t^{(s)} p_\theta\left(Y_{ij} = y \mid W^{(s)}\right). \quad (12)$$

This is a well-defined categorical distribution on the finite set $\{0, 1, 2\}$. Its Shannon entropy $H_t^{ij} := H(p_t^{ij}(y))$ is

$$H_t^{ij} = - \sum_{y \in \{0,1,2\}} \widehat{p}_t^{ij}(y) \, \log \widehat{p}_t^{ij}(y). \quad (13)$$

### 6.2. Expected Information Gain

The one-step expected information gain (EIG) for querying $(i, j)$ is the mutual information between the random variables $W$ and $Y_{ij}$ under the current posterior. In other words,

we aim to select a query that is most informative about the DAG structure. It is easy to show (see Section C) that this is equivalent to the Bayesian active learning by disagreement (BALD, Houlsby et al., 2011) objective:

$$\mathrm{EIG}_t(i, j) := H_t^{ij} - \mathbb{E}_{W \sim q_t}[H(p_\theta(Y_{ij} \mid W))]. \quad (14)$$

Under the particle approximation $q_t(W) \approx \sum_s w_t^{(s)} \delta_{W^{(s)}}$, the mutual information becomes

$$\mathrm{EIG}_t(i, j) \approx H_t^{ij} - \sum_{s=1}^{S} w_t^{(s)} H\left(p_\theta(Y_{ij} \mid W^{(s)})\right), \quad (15)$$

where $H\left(p_\theta(Y_{ij} \mid W^{(s)})\right) = -\sum_{y \in \{0,1,2\}} p_\theta(Y_{ij} = y \mid W^{(s)}) \log p_\theta(Y_{ij} = y \mid W^{(s)})$. Estimating the EIG via Equation (15) has a computational advantage over the standard posterior-based EIG. Indeed, rather than considering entropies over the high-dimensional posterior over $W$, we only estimate entropies for a 3-state categorical distribution. Conceptually, this version of the $\mathrm{EIG}_t(i, j)$ compares the entropy of the *marginal predictive* distribution (epistemic + aleatoric uncertainty), and the posterior-averaged *conditional entropy* (pure aleatoric uncertainty of the expert model). Thus, it favors pairs where the former is large and the latter is small. See Section 7.1 for details.

### 6.3. Candidate Screening

Evaluating EIG for all $D(D - 1)$ ordered pairs may be expensive. We therefore restrict attention to a screened set $\mathcal{C}_t$ of highly uncertain pairs. For each $(i, j)$ define the posterior edge-existence marginal

$$p_{ij}^{\mathrm{exist}}(t) = \sum_{s=1}^{S} w_t^{(s)} \mathbf{1}\left[W_{ij}^{(s)} \neq 0\right],$$

and its associated Bernoulli variance

$$u_{ij}(t) = p_{ij}^{\mathrm{exist}}(t)\left(1 - p_{ij}^{\mathrm{exist}}(t)\right). \quad (16)$$

The screening step selects the $k$ pairs with largest $u_{ij}(t)$:

$$\mathcal{C}_t = \text{Top-}k\left\{u_{ij}(t) : i \neq j\right\}. \quad (17)$$

This top-$k$ step is only a computational screening heuristic. The final policy still maximizes the same information-theoretic acquisition function in Equation (15); screening simply avoids evaluating the EIG score on pairs whose posterior edge-existence marginals are less informative.

### 6.4. Acquisition Decision Rule

We now describe our expert-acquisition rule as follows. At round $t$, the next query is chosen by

$$(i_{t+1}, j_{t+1}) \in \arg \max_{(i,j) \in \mathcal{C}_t} \mathrm{Acq}_t(i, j), \quad (18)$$

**Algorithm 1** Main causal preference elicitation (CaPE) algorithm. At each round, the algorithm screens candidate edges, selects a query according to a policy, obtains expert feedback, and updates the posterior distribution over DAGs.

1: **Input:** Observational data $X$, prior samples $\{W^{(s)}\}_{s=1}^{S} \sim q_0(W \mid X)$, number of particles $S$, rounds $T$, expert parameters $\theta$ (e.g. $(\gamma, \beta_{\text{edge}}, \beta_{\text{dir}}, \lambda)$), query policy $\pi$.
2: Set initial weights $w^{(s)} = 1/S, s = 1, \ldots, S$
3: **for** $t = 1$ **to** $T$ **do**
4:     $\mathcal{C}_t \leftarrow \text{CANDIDATESCREEN}(\{W^{(s)}, w^{(s)}\})$
5:     $(i_t, j_t) \leftarrow \text{SELECTQUERY}(\mathcal{C}_t, \pi, \{W^{(s)}, w^{(s)}\}, \theta)$
6:     $y_t \leftarrow \text{QUERYEXPERT}(i_t, j_t, \theta)$
7:     $\psi_t \leftarrow \{y_t, i_t, j_t, W^{(s)}, w^{(s)}, \theta\}$
8:     $\{W^{(s)}, w^{(s)}\} \leftarrow \text{BAYESIANUPDATE}(\psi_t)$
9:     Record posterior marginals and evaluation metrics.
10: **end for**
11: **return** final posterior $q_T(W)$

where the acquisition score is defined as

$$
\text{Acq}_t(i,j) = \begin{cases} \text{EIG}_t(i,j), & \text{EIG policy,} \\ u_{ij}(t), & \text{Uncertainty policy,} \\ \text{Uniform}((i,j) \in \mathcal{C}_t), & \text{Random policy.} \end{cases}
$$

The EIG is estimated using Equation (15) and is the only objective that directly measures the expected reduction in global uncertainty over $W$. The uncertainty policy simply selects the Top-1 candidate from $\mathcal{C}_t$ above and the random policy chooses $(i,j)$ uniformly at random.

### 6.5. Algorithm

Our main causal preference elicitation (CaPE) algorithm is presented in Algorithm 1, with the corresponding subroutines given in Algorithms 2 to 5 in the Appendix. It iteratively refines a posterior distribution over directed acyclic graphs (DAGs) through a sequence of targeted queries to an oracle. Each iteration comprises four modular stages: *candidate screening*, *query selection*, *expert feedback*, and *Bayesian update*. The modular decomposition allows for easy replacement of components such as the query policy or expert model. See Section E for details. We will make our code publicly available upon acceptance.

**Computational complexity.** At each elicitation round, CaPE maintains a particle approximation with $S$ weighted DAGs. The dominant costs are: (i) screening candidate node pairs based on posterior edge marginals, which costs $O(SD^2)$ in the worst case but is followed by selection of a screened set of size $k \ll D^2$; (ii) computing the expected information gain (EIG) for screened pairs, which scales as $O(Sk)$; and (iii) particle reweighting under a single expert response, which costs $O(S)$. Resampling and Metropolis-Hastings rejuvenation are triggered only when the effective sample size falls below a threshold and incur an additional $O(S)$ cost per rejuvenation step. Overall, the per-round time complexity is $O(SD^2 + Sk)$, with memory cost $O(SD^2)$ to store the particle graphs. In practice, computation is dominated by the $O(Sk)$ term due to aggressive screening ($k \ll D^2$), and all particle-wise operations are embarrassingly parallel. Full details and empirical scaling behavior are provided in Section E.1.

## 7. Conceptual and Theoretical Insights

### 7.1. Uncertainty Reduction and Posterior Contraction

Our method is underpinned by the maximization of the EIG, as defined in Equation (14), where the entropy $H_t^{ij}$ captures the total uncertainty in the predicted expert response. The term $\mathbb{E}_{W \sim q_t}[H(p_\theta(Y_{ij} \mid W))]$ captures the intrinsic aleatoric noise of the expert model. Subtracting the two yields the component of predictive uncertainty that is *reducible* by learning $W$.

Thus, $\text{EIG}_t(i,j)$ is large exactly when: (i) different candidate graphs $W^{(s)}$ make conflicting yet confident predictions for $Y_{ij}$ (*high epistemic uncertainty*); and (ii) for any fixed $W$, the expert model is not inherently noisy (*low aleatoric uncertainty*). In this sense, EIG prioritizes pairs $(i,j)$ for which observing $Y_{ij}$ is expected to most reduce global posterior uncertainty about the underlying DAG. A different view of this uncertainty-reduction principle is that our method maximizes expected posterior change, before and after observing $Y_{ij}$. More concretely, we can show that the EIG is an expected posterior contraction in the KL sense.

**Proposition 7.1** (EIG and expected KL contraction). *Fix $(i,j)$ and let $q_t(W)$ denote the current posterior. For each possible label $y \in \{0, 1, 2\}$, define the updated posterior after hypothetically observing $Y_{ij} = y$ by*

$$
q_t(W \mid Y_{ij} = y) = \frac{q_t(W)\, p_\theta(Y_{ij} = y \mid W)}{\widehat{p}_t^{ij}(y)},
$$

*where $\widehat{p}_t^{ij}(y)$ is the posterior predictive probability of $Y_{ij} = y$. Then the EIG can be written as*

$$
EIG_t(i,j) = \sum_y \widehat{p}_t^{ij}(y)\, \text{KL}\big(q_t(W \mid Y_{ij} = y) \,\|\, q_t(W)\big).
$$

thus, choosing $(i,j)$ to maximize $\text{EIG}_t(i,j)$ is equivalent to choosing the query that maximizes the expected KL divergence between the current posterior and the posterior after observing the expert's answer. See Section D.2 for details and proof. As it turns out, the EIG also has an interesting geometric interpretation. In short, the EIG quantifies the degree of *posterior disagreement* about the expert's answer in the probability simplex. See Section D.3 for details.

## 7.2. Identifiability Results

While reduction in posterior uncertainty results are interesting and useful in their own right, it is important to analyze whether our method can converge to the true underlying causal graph $W^*$, if this exists. Below we provide an answer under a setting with non-adversarial feedback, i.e., the expert assigns strictly higher probability to the correct orientation than to any incorrect one.

**Theorem 7.1** (Identifiability under non-adversarial expert feedback). *Let $W^\star$ be the true weighted DAG generating data $X$. Assume an expert answers queries $(i, j)$ according to a likelihood $p_\theta(y \mid W^\star)$ with $\theta > 0$ such that for every pair $(i, j)$,*

$$p_\theta(y_{ij}^\star \mid W^\star) > p_\theta(y \mid W^\star) \quad \forall y \neq y_{ij}^\star,$$

*where $y_{ij}^\star$ denotes the correct edge status among $\{i \to j, j \to i, \varnothing\}$. Suppose further that each ambiguous pair $(i, j)$ is queried infinitely often as $t \to \infty$. Then the sequence of posteriors*

$$q_t(W \mid X, \mathcal{D}_{1:t}) \propto q_{t-1}(W \mid X, \mathcal{D}_{1:t-1}) \, p_\theta(y_t \mid W)$$

*converges almost surely to a distribution concentrated on $W^\star$, up to the Markov equivalence class determined by $X$.*

Intuitively, each expert answer provides a consistent likelihood factor favoring $W^\star$ over alternatives. By the law of large numbers, repeated queries drive the posterior odds of any incorrect edge orientation to zero, provided every ambiguous edge is queried infinitely often. In finite time, the EIG prioritizes precisely those ambiguous edges, thus accelerating convergence. Details, proof and additional concentration results can be found in Section D.4.

The identifiability result should be interpreted as a consistency guarantee for the elicitation mechanism under idealized coverage and non-adversarial feedback assumptions, rather than as a complete finite-budget characterization of the adaptive policy. Finite-budget behavior is governed by the one-step EIG criterion and is assessed empirically below; the finite-time concentration result in Appendix D.4 provides supporting intuition through a KL-margin condition.

## 8. Experiments

We evaluate our CaPE framework on synthetic data, the Sachs protein signaling network dataset (Sachs et al., 2005) and the CausalBench human gene perturbation benchmark (K562 cell line, Chevalley et al., 2025).

**Baselines.** We compare CaPE against random querying (RND), uncertainty sampling based on marginal edge entropy (UNC), as described in Section 6.4, and a strong non-adaptive information-theoretic baseline that ranks queries by

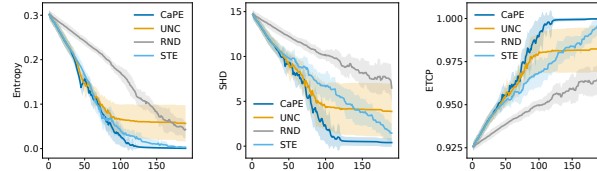

*Figure 1.* Synthetic-data results comparing expert query strategies. Average predictive entropy (Entropy ↓), structural Hamming distance (SHD ↓) to the ground-truth graph, and expected true class probability (ETCP ↑). The means ± one standard deviations (as shaded areas) across 10 replications are reported.

expected information gain computed once from the initial posterior (STE, see Section G.1)[2]. Additionally, towards the end of this section, we evaluate our framework when using a much more elaborate prior obtained from training a DAG-generating GFlowNet (Deleu et al., 2022).

**Evaluation metrics.** We evaluate performance throughout the elicitation process using both uncertainty and structural measures. Here we report average predictive entropy over queried edge relations (Entropy), expected true-class probability (ETCP), and structural Hamming distance (SHD). On CausalBench we report the area under the precision-recall curve AUPRC and top-K precision. This mix is deliberate: entropy and ETCP directly evaluate posterior concentration, while AUPRC and Top-$K$ precision are more informative than AUROC for sparse directed effect graphs (Karimi Mamaghan et al., 2024). For more details of these metrics and others (such as orientation F1 mentioned in Section 8.3), see Section G.2.

### 8.1. Synthetic DAG Experiment

Here we use a controlled synthetic causal discovery task in which both prior samples and expert responses are generated from a known ground-truth DAG. The experiment is designed to test whether expert feedback, when combined with Bayesian active learning, drives posterior concentration towards the true graph. The true graphs are generated from an Erdős–Rényi with edge probability $p_{\text{true}} = 0.5$, $D = 20$ and the prior samples are given by noisy versions of this graph. See Section G.4 for full details.

Figure 1 shows the results across different rounds where we see that, as expected, all algorithms improve the metrics as they receive more expert feedback. Importantly, CaPE outperforms competing approaches in entropy reduction, SHD to the ground-truth graph and probability mass assigned to this ground truth (as given by the ETCP).

---

[2]Code for da Silva et al. (2025) was unavailable at time of writing, and so we could not compare to the method.

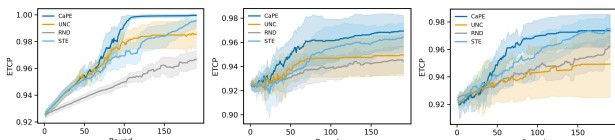

*Figure 2.* Robustness to expert-model misspecification. the ETCP ($\uparrow$) under several non-adversarial departures from the assumed expert model: heterogeneous reliability (left), directional bias (middle) and very noisy (right) regimes. All regimes and metrics in Appendix H.2.

## 8.2. Robustness to Expert Misspecification

Real experts are unlikely to follow the assumed likelihood exactly. We therefore evaluate CaPE under several misspecified oracle regimes while keeping the inference model fixed at the same settings as in Section 8.1. The regimes include pairwise heterogeneous reliability, systematic directional bias, abstention bias, and substantially noisier experts. See Section H.2 for full details and results.

Figure 2 shows three of these regimes, where CaPE remains better than or comparable to the baselines, indicating that the gains are not an artifact of a matched synthetic oracle. The most important qualitative observation is that CaPE is robust to structured expert-response deviations from the assumed likelihood. These experiments are not intended to cover adversarial experts; rather, they test realistic non-adversarial failures of calibration and consistency.

## 8.3. Sachs Protein Signaling (Observational Only)

The protein signaling dataset of Sachs et al. (2005), consists of flow-cytometry measurements of $D = 11$ signaling proteins in human T cells. Following standard practice, we use the observational-only subset of $N = 853$ observations and compare learned graphs to the 11-node, 17-edge reference network reported in the original study. The initial posterior is approximated by $S = 500$ DAG particles generated via bootstrap linear regression with bounded parent sets. Expert feedback is modeled as three-way categorical judgments on edge existence and orientation using the reference network. Posterior inference is performed by importance reweighting with ESS-based resampling and lightweight MCMC rejuvenation See Section G.5 for more details.

Figure 3 provides a quantitative evaluation of the methods where, as before, we see that CaPE outperforms competing approaches on uncertainty reduction, structural metrics and predictive edge probabilities. In particular, after $T = 40$ queries, CaPE reduces SHD by approximately 14 edges relative to the initial posterior and improves orientation F1 by more than 0.38 on average (see Table 3 in Appendix). The final SHD and F1 values obtained by CaPE are consistent with, and in several cases improve upon, those reported for strong score-based and continuous optimization methods

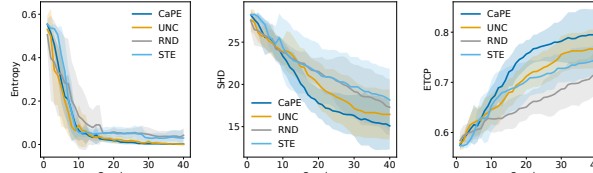

*Figure 3.* Results on Sachs observational-only benchmark: Average predictive entropy (Entropy $\downarrow$), structural Hamming distance (SHD $\downarrow$) to the ground-truth graph, and expected true class probability (ETCP $\uparrow$). The means $\pm$ one standard deviations (as shaded areas) across 10 replications are reported.

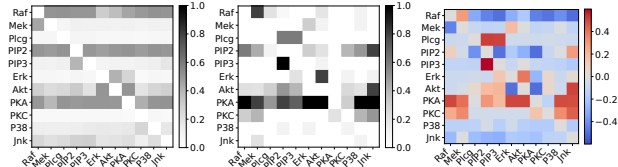

*Figure 4.* Posterior edge probabilities on Sachs (observational-only): (left) initial observational posterior $q_0$, (middle) posterior after $T{=}40$ EIG-selected expert queries, (right) difference between final and initial posteriors. Targeted querying sharpens posterior beliefs over directed edges.

in prior work (Thompson et al., 2025; Zheng et al., 2018). We emphasize that these improvements are achieved using observational data only, without access to the interventional samples available in the original Sachs study[3].

Finally, as a qualitative analysis, Figure 4 visualizes the posterior edge probabilities before and after $T{=}40$ EIG queries, illustrating how a small number of targeted expert judgments sharply concentrates posterior mass over the causal structure.

## 8.4. CausalBench Human Gene Perturbation

We evaluate our method on the CRISPR perturbation dataset from CausalBench (Chevalley et al., 2025), consisting of gene expression measurements in the K562 human leukemia cell line following single-gene perturbations. We construct a 50-node instance by selecting the genes with highest marginal variance in the observational subset, yielding $N = 8{,}553$ samples and $D = 50$ variables. An oracle effect graph is derived exclusively from interventional data. The initial observational posterior is represented similarly to that in Sachs. Because the oracle defines an effect graph rather than a ground-truth DAG, we evaluate posterior edge marginals using directed AUPRC and Top-$K$ precision against the oracle. We emphasize AUPRC rather than AUROC because directed effect graphs are sparse, making

---

[3]We note that some previous work (e.g., Thompson et al., 2025) report using the combined 7,466 samples, in contrast to our *observational-only* study having only 853 observations.

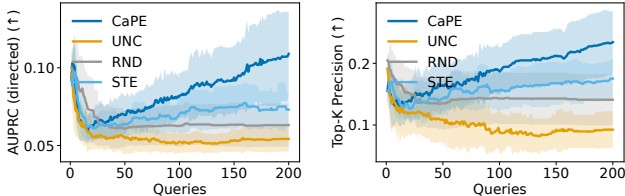

*Figure 5.* Results on the CausalBench K562 50-gene benchmark. We report directed AUPRC (left) and Top-$K$ precision (right, with $K$ equal to the number of oracle edges) as a function of the number of expert queries. Shaded regions indicate $\pm 1$ standard deviation over random seeds.

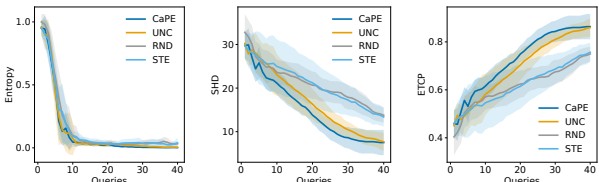

*Figure 6.* Results on Sachs observational-only benchmark with a DAG-GFN prior: Average predictive entropy (Entropy $\downarrow$), structural Hamming distance (SHD $\downarrow$) to the ground-truth graph, and expected true class probability (ETCP $\uparrow$). Shaded regions indicate $\pm 1$ standard deviation over random seeds.

precision–recall behavior more informative for assessing whether posterior mass concentrates on experimentally supported effects. Full details can be found in Section G.6.

Figure 5 show the results on CausalBench. We observe a brief decrease in performance at very small query budgets across all methods, reflecting the fact that early expert feedback is highly local and can temporarily disrupt global ranking metrics before sufficient information accumulates. After that we see that CaPE consistently improves with additional expert queries and achieves the strongest performance at moderate and large query budgets.

The strong static baseline (STE) is competitive at small budgets but saturates quickly and does not benefit from further interaction. In contrast, CaPE exhibits monotonic gains in both AUPRC and Top-$K$ precision, clearly outperforming all baselines by $T \geq 100$. Naive uncertainty sampling (UNC) degrades as the budget grows, underscoring the importance of information-theoretic query selection.

### 8.5. DAG-GFlowNet Prior

To assess the impact of stronger observational priors, we train a DAG-generating GFlowNet (DAG-GFN, Deleu et al., 2022) on Sachs. We adopt a training procedure analogous to that used by da Silva et al. (2025) on ancestral graphs. See Section H.1 for details. We initialize our particle posterior using samples drawn from the trained DAG-GFN and then run our CaPE method with identical settings as before. This

experiment illustrates the modularity of CaPE: the posterior-refinement step does not depend on the particular sampler used to produce the initial DAG particles.

We see in Figure 6 that initializing CaPE with a DAG-GFN prior improves performance relative to the baselines, consistently reducing average predictive entropy and SHD while increasing the expected true class probability (ETCP) across iterations. When compared to Figure 3, we see significant improvements on SHD and ETCP. These results highlight that strong observational priors gracefully complement our causal preference elicitation framework.

## 9. Discussion, Limitations, and Future Work

CaPE highlights a complementary regime for causal discovery in which structural uncertainty is reduced through targeted expert interaction rather than additional experimental data alone. The framework assumes non-adversarial experts: responses may be noisy, heterogeneous, biased, or abstaining, but should still retain information about the underlying graph. In practice, expert judgments may also be correlated or systematically miscalibrated, motivating future work on adaptive and expert-specific reliability models.

CaPE is further limited by the locality of its queries. Each interaction concerns a single variable pair, while the posterior itself lives on a combinatorial DAG space, so resolving global uncertainty may still require multiple rounds of elicitation. Scalability is another challenge: the dominant per-round cost is $O(SD^2 + Sk)$, where $D$ is the number of variables, $S$ the number of samples and $k$ the cardinality of the screening set. This is practical for the regimes studied here ($D \leq 50$) but will require stronger screening, amortization, or sparse priors at larger scales. Moreover, CaPE operates on DAG posteriors and therefore does not explicitly model latent confounding through bidirected edges.

More broadly, expert elicitation and intervention design should be viewed as complementary paradigms. Future systems could jointly allocate budget between querying experts and collecting interventional data, while also addressing deployment challenges such as expert disagreement, query interpretability, and uncertainty communication.

Overall, CaPE provides a modular probabilistic framework for expert-in-the-loop causal discovery, showing that carefully budgeted expert interaction can substantially accelerate posterior concentration across synthetic and real benchmarks, even under several realistic forms of expert-model misspecification.

## Impact Statement

This work contributes to expert-in-the-loop causal discovery by introducing a probabilistic framework for refining

uncertainty over causal structure through targeted expert interaction. Potential applications include scientific domains such as biology and medicine, where interventions may be costly, limited, or ethically constrained. By combining observational evidence with structured expert feedback, such systems may help improve the efficiency and transparency of scientific hypothesis generation.

At the same time, the quality of inferred causal structure depends on the reliability and calibration of expert judgments. Overconfident, systematically biased, or poorly calibrated experts could reinforce incorrect structural assumptions if not properly modeled. More broadly, deploying human-in-the-loop causal systems raises questions about interpretability, uncertainty communication, and appropriate reliance on automated recommendations in scientific decision-making.

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

## A. Structural features

The local direction score combines weight evidence with a structural feature:

$$s_{i \to j}(W) \; = \; g(|W_{ij}|) \; + \; \lambda \, \phi_{i \to j}(W).$$

We consider the following choices for $\phi_{i \to j}(W)$:

- *Posterior log-odds:*

$$\phi_{i \to j}^{\mathrm{odds}} = \log \frac{p_{i \to j}^{t} + \epsilon}{p_{j \to i}^{t} + \epsilon}, \quad p_{i \to j}^{t} = \Pr_{t}(A_{ij} = 1, A_{ji} = 0).$$

- *V-structure support:*

$$\phi_{i \to j}^{\mathrm{coll}} = \frac{1}{D - 2} \sum_{k \notin \{i,j\}} \Pr_{t}\big(i \to j \leftarrow k \text{ is an unshielded collider}\big) - (i \leftrightarrow j).$$

- *Cycle-risk asymmetry:*

$$\phi_{i \to j}^{\mathrm{acyc}} = \Pr_{t}(\text{adding } j \to i \text{ induces a cycle}) - \Pr_{t}(\text{adding } i \to j \text{ induces a cycle}).$$

In practice, we can combine features linearly,

$$\phi_{i \to j}(W) \; = \; \sum_{k} \alpha_k \, \phi_{i \to j}^{(k)}(W),$$

with coefficients $\alpha_k$ learned online by maximizing the cumulative expert log-likelihood, or treated as fixed hyperparameters.

## B. Inference Details

### B.1. Particle Rejuvenation

After resampling, the particle set $\{W^{(s)}\}_{s=1}^{S} \sim q_t(W)$ consists largely of duplicate graphs with equal weights. To restore diversity, we apply a *rejuvenation kernel* $K_t(W' \mid W)$ that leaves the current posterior $q_t(W)$ invariant. Each particle undergoes one or several small Metropolis–Hastings (MH) proposals:

$$W' \sim K_t(\cdot \mid W^{(s)}), \qquad a(W', W^{(s)}) = \min\left(1, \frac{q_t(W') \, K_t(W^{(s)} \mid W')}{q_t(W^{(s)}) \, K_t(W' \mid W^{(s)})}\right),$$

where $a(W', W^{(s)})$ is the MH acceptance probability. If the proposal is accepted, we replace $W^{(s)} \leftarrow W'$; otherwise the particle is retained.

In our implementation, the proposal kernel $K_t$ performs a small random *graph edit*:

$$K_t(W' \mid W) = \begin{cases} \text{flip orientation of a random edge,} \\ \text{add or remove a random edge,} \\ 0 \quad \text{if } W' \text{ is cyclic.} \end{cases}$$

Acyclicity is enforced by rejecting any $W'$ that violates the DAG constraint. The acceptance ratio uses the current posterior density (up to a constant),

$$\log \frac{q_t(W')}{q_t(W)} = \log q_0(W') - \log q_0(W) + \sum_{(i,j,y) \in \mathcal{D}_t} \Big[ \log p_\theta(y \mid W', i, j) - \log p_\theta(y \mid W, i, j) \Big],$$

where $q_0(W)$ is the observational prior and $\mathcal{D}_t$ is the set of expert responses up to round $t$. This MCMC rejuvenation step maintains the target posterior while preventing particle impoverishment.

## C. Expected Information Gain for Query Selection

Here we give a somewhat informal derivation of our EIG expression and a conceptual interpretation. A formal derivation is given in Section D.1. We derive the expected information gain (EIG) for a single expert query about an ordered pair $(i, j)$. All randomness is taken with respect to the current posterior $q_t(W)$ and the expert-likelihood model $p_\theta(Y_{ij} = y \mid W)$ introduced earlier.

### C.1. Mutual-information Definition

For fixed $(i, j)$, the expert-response variable $Y_{ij}$ takes values in the finite alphabet $\{0, 1, 2\}$. The expected information gain of querying $(i, j)$ at round $t$ is the mutual information

$$\text{EIG}_t(i, j) \;=\; I_t(W; Y_{ij}),$$

defined by

$$I_t(W; Y_{ij}) = \mathbb{E}_{W, Y_{ij}} \left[ \log \frac{p(W, Y_{ij})}{p(W)\, p(Y_{ij})} \right].$$

Since the posterior is $p(W) = q_t(W)$ and the joint factorizes as $p(W, Y_{ij}) = q_t(W)\, p_\theta(Y_{ij} \mid W)$, the expression becomes

$$I_t(W; Y_{ij}) = \sum_W q_t(W) \sum_{y \in \{0,1,2\}} p_\theta(y \mid W) \log \frac{p_\theta(y \mid W)}{\widehat{p}_t^{ij}(y)},$$

where the posterior predictive distribution is

$$\widehat{p}_t^{ij}(y) = \sum_W q_t(W)\, p_\theta(Y_{ij} = y \mid W) = \sum_{s=1}^{S} w_t^{(s)} p_\theta(Y_{ij} = y \mid W^{(s)}).$$

### C.2. BALD Entropy Decomposition

Mutual information admits the standard identity

$$I_t(W; Y_{ij}) = H(Y_{ij}) - H(Y_{ij} \mid W),$$

where $H(\cdot)$ denotes Shannon entropy. The marginal entropy of $Y_{ij}$ is

$$H_t^{ij} = H(Y_{ij}) = - \sum_{y \in \{0,1,2\}} \widehat{p}_t^{ij}(y) \log \widehat{p}_t^{ij}(y).$$

The posterior expectation of the conditional entropy is

$$\mathbb{E}_{W \sim q_t}[H(Y_{ij} \mid W)] \approx \sum_{s=1}^{S} w_t^{(s)} \left( - \sum_{y \in \{0,1,2\}} p_\theta(Y_{ij} = y \mid W^{(s)}) \log p_\theta(Y_{ij} = y \mid W^{(s)}) \right).$$

Combining these,

$$\boxed{\text{EIG}_t(i, j) = H_t^{ij} - \sum_{s=1}^{S} w_t^{(s)}\, H\left( p_\theta(Y_{ij} \mid W^{(s)}) \right)}$$

which is exactly the BALD (Bayesian Active Learning by Disagreement) decomposition specialized to the random variable $Y_{ij}$.

### C.3. Interpretation

The entropy $H_t^{ij}$ captures the total uncertainty in the predicted expert response. The term $\mathbb{E}_{W \sim q_t}[H(Y_{ij} \mid W)]$ captures the intrinsic aleatoric noise of the expert model. Subtracting the two yields the component of predictive uncertainty that is *reducible* by learning $W$.

Thus $\text{EIG}_t(i, j)$ is large exactly when:

(i) different candidate graphs $W^{(s)}$ make conflicting yet confident predictions for $Y_{ij}$ (high epistemic uncertainty), and

(ii) for any fixed $W$, the expert model is not inherently noisy (low aleatoric uncertainty).

In this sense, EIG prioritizes pairs $(i, j)$ for which observing $Y_{ij}$ is expected to most reduce global posterior uncertainty about the underlying DAG.

## D. Theory

### D.1. Mutual Information and BALD Decomposition

We first formalize the BALD-style decomposition of the expected information gain for a fixed ordered pair $(i, j)$.

**Lemma D.1** (BALD decomposition for $Y_{ij}$). *Fix an ordered pair $(i, j)$ and let $Y_{ij}$ be the corresponding expert-response variable taking values in $\{0, 1, 2\}$. Let $q_t(W)$ denote the current posterior over weighted DAGs, and let $p_\theta(Y_{ij} = y \mid W)$ be the expert likelihood. Define the posterior predictive distribution*

$$\widehat{p}_t^{ij}(y) = \sum_W q_t(W) \, p_\theta(Y_{ij} = y \mid W), \qquad y \in \{0, 1, 2\}.$$

*Then the mutual information between $W$ and $Y_{ij}$ under $q_t$ satisfies*

$$I_t(W; Y_{ij}) = H\big(\widehat{p}_t^{ij}\big) - \mathbb{E}_{W \sim q_t}\big[H\big(p_\theta(Y_{ij} \mid W)\big)\big],$$

*where $H(\cdot)$ denotes Shannon entropy on the finite set $\{0, 1, 2\}$. In particular, under the particle approximation $q_t(W) \approx \sum_{s=1}^S w_t^{(s)} \delta_{W^{(s)}}(W)$,*

$$I_t(W; Y_{ij}) = H\big(\widehat{p}_t^{ij}\big) - \sum_{s=1}^S w_t^{(s)} H\big(p_\theta(Y_{ij} = \cdot \mid W^{(s)})\big).$$

*Proof.* By definition, the mutual information between $W$ and $Y_{ij}$ is

$$I_t(W; Y_{ij}) = H(Y_{ij}) - H(Y_{ij} \mid W).$$

Since $Y_{ij}$ is supported on the finite alphabet $\{0, 1, 2\}$,

$$H(Y_{ij}) = - \sum_{y \in \{0,1,2\}} \widehat{p}_t^{ij}(y) \log \widehat{p}_t^{ij}(y) = H\big(\widehat{p}_t^{ij}\big).$$

Moreover,

$$H(Y_{ij} \mid W) = \mathbb{E}_{W \sim q_t}\big[H\big(p_\theta(Y_{ij} = \cdot \mid W)\big)\big] = \sum_W q_t(W) \Big(-\sum_y p_\theta(Y_{ij} = y \mid W) \log p_\theta(Y_{ij} = y \mid W)\Big).$$

Substituting these expressions into $I_t(W; Y_{ij}) = H(Y_{ij}) - H(Y_{ij} \mid W)$ yields the claimed identity. Under the particle approximation, the expectation $\mathbb{E}_{W \sim q_t}[\cdot]$ reduces to the finite weighted sum $\sum_{s=1}^S w_t^{(s)}[\cdot]$, which proves the second expression. $\square$

### D.2. EIG as Expected Posterior Contraction

We show that, for a fixed pair $(i, j)$, the expected information gain is exactly the expected Kullback–Leibler (KL) divergence between the current posterior and the posterior after observing $Y_{ij}$.

**Proposition D.1** (Restatement of proposition 7.1, EIG and expected KL contraction). *Fix $(i, j)$ and let $q_t(W)$ denote the current posterior. For each possible label $y \in \{0, 1, 2\}$, define the updated posterior after hypothetically observing $Y_{ij} = y$ by*

$$q_t(W \mid Y_{ij} = y) = \frac{q_t(W) \, p_\theta(Y_{ij} = y \mid W)}{\widehat{p}_t^{ij}(y)},$$

*where $\widehat{p}_t^{ij}(y)$ is the posterior predictive probability of $Y_{ij} = y$. Then the mutual information between $W$ and $Y_{ij}$ can be written as*

$$I_t(W; Y_{ij}) = \mathbb{E}_{Y_{ij}}\big[\mathrm{KL}\big(q_t(W \mid Y_{ij}) \,\|\, q_t(W)\big)\big],$$

*where the expectation is with respect to the marginal $p(Y_{ij} = y) = \widehat{p}_t^{ij}(y)$. Equivalently,*

$$I_t(W; Y_{ij}) = \sum_{y \in \{0,1,2\}} \widehat{p}_t^{ij}(y) \, \mathrm{KL}\big(q_t(W \mid Y_{ij} = y) \,\|\, q_t(W)\big).$$

*Proof.* Starting from the definition of mutual information,

$$I_t(W; Y_{ij}) = \sum_{y \in \{0,1,2\}} \sum_W p(W, Y_{ij} = y) \log \frac{p(W, Y_{ij} = y)}{p(W)\, p(Y_{ij} = y)}.$$

Using $p(W) = q_t(W)$ and $p(W, Y_{ij} = y) = q_t(W)\, p_\theta(Y_{ij} = y \mid W)$, we have

$$I_t(W; Y_{ij}) = \sum_y \sum_W q_t(W)\, p_\theta(Y_{ij} = y \mid W) \log \frac{q_t(W)\, p_\theta(Y_{ij} = y \mid W)}{q_t(W)\, \widehat{p}_t^{ij}(y)}.$$

Simplifying,

$$I_t(W; Y_{ij}) = \sum_y \sum_W q_t(W)\, p_\theta(Y_{ij} = y \mid W) \log \frac{p_\theta(Y_{ij} = y \mid W)}{\widehat{p}_t^{ij}(y)}.$$

Rewriting the joint as $p(W, Y_{ij} = y) = p(Y_{ij} = y)\, q_t(W \mid Y_{ij} = y) = \widehat{p}_t^{ij}(y)\, q_t(W \mid Y_{ij} = y)$, we obtain

$$I_t(W; Y_{ij}) = \sum_y \widehat{p}_t^{ij}(y) \sum_W q_t(W \mid Y_{ij} = y) \log \frac{q_t(W \mid Y_{ij} = y)}{q_t(W)},$$

which is

$$I_t(W; Y_{ij}) = \sum_y \widehat{p}_t^{ij}(y) \, \mathrm{KL}\big(q_t(W \mid Y_{ij} = y) \,\|\, q_t(\cdot)\big).$$

This is exactly the expected KL divergence between the updated and current posteriors, with expectation taken over the predictive distribution of $Y_{ij}$. $\qquad\square$

*Remark* D.1 (Query selection as maximizing expected posterior change). Proposition 7.1 shows that, for a fixed round $t$, choosing $(i, j)$ to maximize $\mathrm{EIG}_t(i, j)$ is equivalent to choosing the query that maximizes the expected KL divergence between the current posterior and the posterior after observing the expert's answer. In this sense, EIG is the acquisition rule that maximizes the expected amount by which the query will update our beliefs about the DAG.

### D.3. Geometric Interpretation in the Probability Simplex

It is sometimes useful to view the expert likelihoods and predictive distribution as points in the probability simplex over the finite alphabet $\{0, 1, 2\}$.

For a fixed pair $(i, j)$ and graph $W$, define the likelihood vector

$$\mathbf{p}^{ij}(W) = \big(p_\theta(Y_{ij} = 0 \mid W), p_\theta(Y_{ij} = 1 \mid W), p_\theta(Y_{ij} = 2 \mid W)\big) \in \Delta^2,$$

where $\Delta^2$ is the 2-simplex of categorical distributions on $\{0, 1, 2\}$. The posterior predictive distribution is the mixture

$$\widehat{\mathbf{p}}_t^{ij} = \sum_W q_t(W)\, \mathbf{p}^{ij}(W) \in \Delta^2.$$

The following alternative expression for the mutual information makes this geometry explicit.

**Lemma D.2** (Mixture form of $I_t(W; Y_{ij})$). *With notation as above,*

$$I_t(W; Y_{ij}) = \sum_W q_t(W) \, \mathrm{KL}\big(\mathbf{p}^{ij}(W) \,\big\|\, \widehat{\mathbf{p}}_t^{ij}\big),$$

*that is, the mutual information is the posterior-weighted average KL divergence between each likelihood vector $\mathbf{p}^{ij}(W)$ and the mixture $\widehat{\mathbf{p}}_t^{ij}$.*

*Proof.* Starting again from the definition

$$I_t(W; Y_{ij}) = \sum_W q_t(W) \sum_y p_\theta(Y_{ij} = y \mid W) \log \frac{p_\theta(Y_{ij} = y \mid W)}{\widehat{p}_t^{ij}(y)},$$

we recognize, for each fixed $W$, the inner sum as

$$\sum_y p_\theta(Y_{ij} = y \mid W) \log \frac{p_\theta(Y_{ij} = y \mid W)}{\widehat{p}_t^{ij}(y)} = \mathrm{KL}\big(\mathbf{p}^{ij}(W) \,\big\|\, \widehat{\mathbf{p}}_t^{ij}\big).$$

Substituting this back into the expression for $I_t(W; Y_{ij})$ yields

$$I_t(W; Y_{ij}) = \sum_W q_t(W) \, \mathrm{KL}\big(\mathbf{p}^{ij}(W) \,\big\|\, \widehat{\mathbf{p}}_t^{ij}\big),$$

which proves the claim. $\qquad\square$

*Remark* D.2 (EIG as disagreement in the simplex). Lemma D.2 shows that $I_t(W; Y_{ij})$ measures how widely spread the likelihood vectors $\mathbf{p}^{ij}(W)$ are around their mixture $\widehat{\mathbf{p}}_t^{ij}$ in the probability simplex. If all candidate graphs $W$ agree on the distribution of $Y_{ij}$, then $\mathbf{p}^{ij}(W) = \widehat{\mathbf{p}}_t^{ij}$ for all $W$ and $I_t(W; Y_{ij}) = 0$. Conversely, when different graphs induce sharply different $\mathbf{p}^{ij}(W)$, the KL divergences to the mixture become large, and so does the expected information gain. In this sense, EIG quantifies the degree of *posterior disagreement* about the expert's answer in the probability simplex.

### D.4. Identifiability and Finite-Time Concentration Results

**Proposition (informal).** Suppose the true causal graph $W^\star$ generates data $X$ and an expert that answers queries $(i, j)$ according to the likelihood $p_\theta(y \mid W^\star)$ with hyperparameters $\theta$. Then, under non-adversarial feedback (i.e., the expert assigns strictly higher probability to the correct orientation than to any incorrect one), the sequence of Bayesian updates

$$q_t(W \mid X, \mathcal{D}_{1:t}) \propto q_{t-1}(W \mid X, \mathcal{D}_{1:t-1}) \, p_\theta(y_t \mid W)$$

converges in the limit $t \to \infty$ to a posterior concentrated on $W^\star$ (up to Markov equivalence if $X$ alone does not distinguish orientations).

*Sketch.* Each expert answer provides a consistent likelihood factor favoring $W^\star$ over alternatives. By the law of large numbers, repeated queries drive the posterior odds of any incorrect edge orientation to zero, provided every ambiguous edge is queried infinitely often. In finite time, Expected Information Gain prioritizes precisely those ambiguous edges, thus accelerating convergence.

**Theorem D.1** (Identifiability under non-adversarial expert feedback). *Let $W^\star$ be the true weighted DAG generating data $X$. Assume an expert answers queries $(i, j)$ according to a likelihood $p_\theta(y \mid W^\star)$ such that for every pair $(i, j)$,*

$$p_\theta(y_{ij}^\star \mid W^\star) > p_\theta(y \mid W^\star) \quad \forall y \neq y_{ij}^\star,$$

*where $y_{ij}^\star$ denotes the correct edge status among $\{\, i \to j, \; j \to i, \; \varnothing \,\}$. Suppose further that each ambiguous pair $(i, j)$ is queried infinitely often as $t \to \infty$. Then the sequence of posteriors*

$$q_t(W \mid X, \mathcal{D}_{1:t}) \propto q_{t-1}(W \mid X, \mathcal{D}_{1:t-1}) \, p_\theta(y_t \mid W)$$

*converges almost surely to a distribution concentrated on $W^\star$, up to the Markov equivalence class determined by $X$.*

*Proof.* Define the posterior odds for any alternative graph $W \neq W^\star$:

$$R_t(W) = \frac{q_t(W \mid X, \mathcal{D}_{1:t})}{q_t(W^\star \mid X, \mathcal{D}_{1:t})}.$$

By Bayes' rule,

$$R_t(W) = R_{t-1}(W) \frac{p_\theta(y_t \mid W)}{p_\theta(y_t \mid W^\star)}.$$

Taking logs and summing,

$$\log R_t(W) = \log R_0(W) + \sum_{\tau=1}^{t} \log \frac{p_\theta(y_\tau \mid W)}{p_\theta(y_\tau \mid W^\star)}.$$

Under the assumption that queries include each ambiguous pair infinitely often and that the expert is non-adversarial, the expected log-ratio for any incorrect $W$ is strictly negative:

$$\mathbb{E}_{y \sim p_\theta(\cdot \mid W^\star)} \left[ \log \frac{p_\theta(y \mid W)}{p_\theta(y \mid W^\star)} \right] < 0.$$

By the strong law of large numbers, the empirical averages converge almost surely to their expectations. Thus $\log R_t(W) \to -\infty$ almost surely, implying $R_t(W) \to 0$. Consequently, the posterior mass concentrates on $W^\star$, up to edges that remain indistinguishable within the Markov equivalence class determined by $X$. □

**Corollary D.1** (Finite-time concentration at exponential rate). *Adopt the setting of the theorem, and fix any incorrect $W \neq W^\star$. Let the expert feedback for a queried pair $(i, j)$ be drawn from $p_\theta(\cdot \mid W^\star)$ whenever $(i, j)$ is asked. Define the one-step log-likelihood ratio*

$$Z_\tau^{(W)} = \log \frac{p_\theta(y_\tau \mid W)}{p_\theta(y_\tau \mid W^\star)}.$$

*Assume:*

1. ***KL margin:*** *There exists*

$$\Delta(W) = \mathrm{KL}\big(p_\theta(\cdot \mid W^\star) \,\|\, p_\theta(\cdot \mid W)\big) > 0.$$

2. ***Bounded increments:*** $Z_\tau^{(W)} \in [-L, L]$ *almost surely for some $L < \infty$.*

3. ***Coverage of distinguishing queries:*** *There is $\eta \in (0, 1]$ such that among the first $T$ rounds, at least $N_T(W) \geq \eta T$ of the queried pairs are* distinguishing *for $W$ (i.e., they would yield different answer distributions under $W^\star$ vs. $W$), and on those rounds the law of $Z_\tau^{(W)}$ is i.i.d. with mean $-\Delta(W)$.*

*Then for any $\alpha \in (0, 1)$, with probability at least $1 - \alpha$,*

$$\log R_T(W) \leq \log R_0(W) - N_T(W)\,\Delta(W) + N_T(W)\,\varepsilon_T,$$

*where*

$$\varepsilon_T = L\sqrt{\frac{2\log(1/\alpha)}{N_T(W)}}.$$

*In particular,*

$$R_T(W) \leq R_0(W) \exp\Big( -N_T(W)\,(\Delta(W) - \varepsilon_T) \Big).$$

*Moreover, letting $\neg\mathcal{W} = \{W \neq W^\star\}$ and $R_0^\Sigma = \sum_{W \in \neg\mathcal{W}} R_0(W)$, a union bound yields that with probability at least $1 - \alpha$,*

$$1 - q_T(W^\star \mid X, \mathcal{D}_{1:T}) = \sum_{W \in \neg\mathcal{W}} q_T(W \mid X, \mathcal{D}_{1:T}) \leq R_0^\Sigma \exp\Big( -\eta T\,(\Delta_{\min} - \varepsilon_T^{\min}) \Big),$$

*where $\Delta_{\min} = \min_{W \in \neg\mathcal{W}} \Delta(W)$ and $\varepsilon_T^{\min} = L\sqrt{2\log(|\neg\mathcal{W}|/\alpha)/(\eta T)}$.*

*Proof.* Write the posterior odds recursion $R_T(W) = R_0(W) \prod_{\tau=1}^{T} \exp(Z_\tau^{(W)})$. By assumption (3), only the $N_T(W)$ distinguishing rounds change $R_T(W)$ in expectation; the rest have zero mean increment (or can be absorbed by redefining $N_T$). Thus

$$\log R_T(W) = \log R_0(W) + \sum_{\tau \in \mathcal{I}_T(W)} Z_\tau^{(W)},$$

where $\mathcal{I}_T(W)$ indexes the $N_T(W)$ distinguishing rounds. By (1), $\mathbb{E}[Z_\tau^{(W)}] = -\Delta(W)$; by (2) and Hoeffding's inequality, for any $\alpha \in (0,1)$, with probability $\geq 1 - \alpha$,

$$\frac{1}{N_T(W)} \sum_{\tau \in \mathcal{I}_T(W)} Z_\tau^{(W)} \leq -\Delta(W) + L\sqrt{\frac{2 \log(1/\alpha)}{N_T(W)}}.$$

Multiplying by $N_T(W)$ and adding $\log R_0(W)$ yields the stated bound on $\log R_T(W)$, hence the exponential bound on $R_T(W)$. For the total posterior error, note $1 - q_T(W^\star) = \sum_{W \neq W^\star} q_T(W) \leq \sum_{W \neq W^\star} R_T(W)$. Apply the previous bound for each $W$ and union bound over $\neg\mathcal{W}$, taking $N_T(W) \geq \eta T$ and the worst-case margin $\Delta_{\min}$, to obtain the final inequality with $\varepsilon_T^{\min} = L\sqrt{2 \log(|\neg\mathcal{W}|/\alpha)/(\eta T)}$. $\qquad \square$

**Notes**

- The decay rate is governed by the KL margin $\Delta(W)$. Larger gaps mean faster collapse of wrong graphs' odds.

- The coverage parameter $\eta$ connects to the query policy. If the select pairs by EIG, $\eta$ is typically bounded away from zero on ambiguous edges; if needed, we can enforce it by exploration (e.g., $\varepsilon$-greedy over pairs).

- The $L$ bound is mild: it holds whenever likelihoods are bounded away from 0 and 1, which we can ensure by a small floor on class probabilities in the 3-way softmax.

**Lemma D.3** (Edge-wise decomposition of the KL margin). *Fix an incorrect graph $W \neq W^\star$. Let the query policy select pairs $(i,j)$ according to a (possibly time-averaged) distribution $\pi = \{\pi_{ij}\}$ supported on pairs for which $p_\theta(\cdot \mid W^\star, i, j) \neq p_\theta(\cdot \mid W, i, j)$. Then the per-round KL margin in Corollary 1 satisfies*

$$\Delta(W) = \sum_{(i,j)} \pi_{ij} \, \text{KL}\Big(p_\theta(\cdot \mid W^\star, i, j) \,\|\, p_\theta(\cdot \mid W, i, j)\Big),$$

*where $p_\theta(\cdot \mid W, i, j)$ denotes the 3-way softmax likelihood over $\{i \to j, \, j \to i, \, \varnothing\}$ for the specific pair $(i,j)$ under graph $W$.*

*Proof.* By definition, $\Delta(W) = \text{KL}(p_\theta(\cdot \mid W^\star) \,\|\, p_\theta(\cdot \mid W))$ for the *one-step* expert channel used in the corollary. If the channel is implemented by first sampling a pair $(i,j) \sim \pi$ and then sampling an answer $y$ from the pair-specific distribution $p_\theta(\cdot \mid W, i, j)$, the overall law is the mixture $p_\theta(y, (i,j) \mid W) = \pi_{ij} \, p_\theta(y \mid W, i, j)$. KL between two such mixtures with the *same* mixing weights $\pi$ decomposes additively:

$$\text{KL}\Big(\pi_{ij} p_\theta(\cdot \mid W^\star, i, j) \,\|\, \pi_{ij} p_\theta(\cdot \mid W, i, j)\Big) = \sum_{(i,j)} \pi_{ij} \, \text{KL}\Big(p_\theta(\cdot \mid W^\star, i, j) \,\|\, p_\theta(\cdot \mid W, i, j)\Big),$$

which is the claimed identity. $\qquad \square$

**Corollary D.2** (Uniform-frequency lower bound). *If the query policy guarantees a uniform lower bound $\pi_{ij} \geq \rho$ for all distinguishing pairs $(i,j)$, then*

$$\Delta(W) \geq \rho \sum_{(i,j) \text{ dist.}} \text{KL}\Big(p_\theta(\cdot \mid W^\star, i, j) \,\|\, p_\theta(\cdot \mid W, i, j)\Big) \geq \rho \, |\mathcal{E}_{\text{dist}}(W)| \, \kappa_{\min}(W),$$

*where $\mathcal{E}_{\text{dist}}(W)$ is the set of pairs for which the two pairwise channels differ, and $\kappa_{\min}(W) = \min_{(i,j) \in \mathcal{E}_{\text{dist}}(W)} \text{KL}\big(p_\theta(\cdot \mid W^\star, i, j) \,\|\, p_\theta(\cdot \mid W, i, j)\big)$.*

**Corollary D.3** (Pinsker-type bound via total variation). *For any pair $(i,j)$ let $V_{ij}(W) = \frac{1}{2}\|p_\theta(\cdot \mid W^\star, i, j) - p_\theta(\cdot \mid W, i, j)\|_1$ be the total variation distance. By Pinsker's inequality,*

$$\Delta(W) = \sum_{(i,j)} \pi_{ij} \, \mathrm{KL}\Big(p_\theta(\cdot \mid W^\star, i, j) \,\big\|\, p_\theta(\cdot \mid W, i, j)\Big) \geq \frac{1}{2\ln 2} \sum_{(i,j)} \pi_{ij} \, V_{ij}(W)^2.$$

*In particular, if $V_{ij}(W) \geq v_0$ on all distinguishing pairs and $\sum_{(i,j)} \pi_{ij} \mathbf{1}[(i,j) \text{ dist.}] \geq \eta$, then $\Delta(W) \geq \frac{\eta}{2\ln 2} v_0^2$.*

*Remark* D.3 (From logits to pairwise divergence). In the 3-class softmax model, the pairwise channel is $p_\theta(\cdot \mid W, i, j) = \mathrm{softmax}(z(W, i, j))$ with logits $z(W, i, j) = \big(\beta s_{i \to j}(W), \beta s_{j \to i}(W), \beta_0\big)$. Let $u_{ij}(W) = z(W, i, j) - z(W^\star, i, j)$ be the logit perturbation. Then $V_{ij}(W)$ (hence the KL term) is controlled by $\|u_{ij}(W)\|$. In particular, since the softmax map is Lipschitz on $\mathbb{R}^3$, there exists $L_{\mathrm{sm}} \in (0, 1)$ such that $V_{ij}(W) \leq L_{\mathrm{sm}} \|u_{ij}(W)\|_2$. Thus Pinsker yields

$$\mathrm{KL}\Big(p_\theta(\cdot \mid W^\star, i, j) \,\big\|\, p_\theta(\cdot \mid W, i, j)\Big) \geq \frac{L_{\mathrm{sm}}^2}{2\ln 2} \|u_{ij}(W)\|_2^2,$$

linking the edge-wise KL margin to squared differences in the underlying direction scores (scaled by $\beta$).

# E. Algorithms

Our main causal preference elicitation (CaPE) algorithm is presented in Algorithm 1 and we described the corresponding subrouines below.

**Candidate screening (Algorithm 2) .** Given the current posterior particle set, we compute for every ordered node pair $(i, j)$ the posterior probability $p_{ij}$ that an edge exists. The quantity $u_{ij} = p_{ij}(1 - p_{ij})$ measures edge-level uncertainty. The screening stage returns the $k$ pairs with highest uncertainty, serving as a candidate pool for the next query.

**Query selection (Algorithm 3).** From the screened set of pairs, a selection policy $\pi \in \{\text{EIG}, \text{UNCERTAINTY}, \text{RANDOM}\}$ determines which pair to query. The expected information gain (EIG) policy computes, for each candidate pair $(i, j)$, the mutual information $I_t(W; Y_{ij})$ between the current posterior over graphs and the yet-unobserved entry $Y_{ij}$ efficiently, as described in Section 6.2.

**Expert feedback (Algorithm 4).** In reality, we have an oracle/expert that gives us feedback. However, in our synthetic experiments we instead simulate an expert using a ground-truth weighted DAG $W^\star$. Once a pair $(i_t, j_t)$ is chosen, the expert provides a label $y_t \in \{0, 1, 2\}$ interpreted as $Y_{i_t j_t}^{(t)} = y_t$ and corresponding to $\{j_t \to i_t, \ i_t \to j_t, \ \text{no direct edge}\}$, drawn from the three-way likelihood $p_\theta(Y_{i_t j_t} = y_t \mid W^\star)$.

**Bayesian update (Algorithm 5).** After receiving the expert response $y_t$, each particle weight is multiplied by the corresponding likelihood value $p_\theta(Y_{i_t j_t} = y_t \mid W^{(s)})$ and renormalized (Algorithm 5). If the effective sample size (ESS) falls below a threshold $\delta_s S$, the particle set is resampled to prevent weight degeneracy. The updated posterior then serves as input to the next round. Across rounds, the posterior sharpens around the true DAG as the system selects increasingly informative queries.

**Particle rejuvenation (Algorithm 6).** Following resampling, we apply a lightweight rejuvenation kernel that randomly perturbs a subset of DAGs by adding, deleting, or flipping edges. Proposals that break acyclicity are rejected, and accepted moves follow a Metropolis–Hastings ratio under the current posterior. This step restores diversity among particles and mitigates degeneracy as the posterior concentrates.

## E.1. Computational Complexity

We analyze the computational complexity of CaPE in terms of the number of variables $D$, the number of posterior particles $S$, the query budget $T$, and the screening budget $k$.

**Posterior representation and updates.** The posterior over DAGs is represented by a weighted particle approximation $\{(W^{(s)}, w^{(s)})\}_{s=1}^S$. At each iteration, incorporating an expert response for a queried pair $(i, j)$ requires evaluating the expert likelihood for that pair across all particles. This costs $\mathcal{O}(S)$ per update, since the likelihood depends only on local edge information in each DAG. Weight normalization and effective sample size (ESS) computation are also $\mathcal{O}(S)$.

When ESS falls below a threshold, multinomial resampling is performed in $\mathcal{O}(S)$ time. If rejuvenation is enabled, each Metropolis-Hastings step proposes local graph edits (edge additions, deletions, or reversals) for a subset of particles. Each

**Algorithm 2** Candidate Screening by Posterior Uncertainty. Computes uncertainty $u_{ij} = p_{ij}(1 - p_{ij})$ for each edge and returns the $k$ most uncertain pairs as candidates.

1: **function** CANDIDATESCREEN($\{W^{(s)}, w^{(s)}\}$)
2:   **for all** pairs $(i, j)$, $i \neq j$ **do**
3:     $p_{ij} \leftarrow \sum_s w^{(s)} \mathbf{1}\{W_{ij}^{(s)} \neq 0\}$
4:     $u_{ij} \leftarrow p_{ij}(1 - p_{ij})$
5:   **end for**
6:   **return** top-$k$ pairs with largest $u_{ij}$
7: **end function**

---

**Algorithm 3** Query Selection Policy. Chooses which candidate pair to ask about. Depending on the policy, selects uniformly at random, takes the most uncertain screened pair, or maximizes expected information gain (EIG).

1: **function** SELECTQUERY($\mathcal{C}, \pi, \{W^{(s)}, w^{(s)}\}, \theta$)
2:   **if** $\pi = $ RANDOM **then**
3:     Choose $(i, j)$ uniformly at random from $\mathcal{C}$
4:   **else if** $\pi = $ UNCERTAINTY **then**
5:     ▷ $\mathcal{C}$ is assumed sorted by decreasing $u_{ij}$
6:     Choose $(i, j)$ as the first element of $\mathcal{C}$
7:   **else if** $\pi = $ EIG **then**
8:     **for all** $(i, j) \in \mathcal{C}$ **do**
9:       Compute $\text{EIG}_t(i, j) = I_t(W; Y_{ij})$ using the current particles and $p_\theta$ (cf. Section C).
10:     **end for**
11:     Choose $(i, j)$ maximizing $\text{EIG}_t(i, j)$
12:   **end if**
13:   **return** $(i, j)$
14: **end function**

---

proposal requires an acyclicity check, implemented via reachability tests, which costs $\mathcal{O}(D^2)$ in the worst case. With a fixed number of rejuvenation steps $R$, the total rejuvenation cost per resampling event is $\mathcal{O}(RSD^2)$.

**Candidate screening.** At each iteration, we compute posterior edge marginals by aggregating particle weights, which requires $\mathcal{O}(SD^2)$ operations. From these marginals, we compute an uncertainty score for each unordered node pair and select the top-$k$ most uncertain pairs. This screening step costs $\mathcal{O}(D^2)$ to compute uncertainties and $\mathcal{O}(D^2 \log k)$ to select the top candidates, and is dominated by the marginal computation.

**Query selection.** For uncertainty-based or random policies, selecting a query from the screened set costs $\mathcal{O}(1)$. For the EIG policy, we compute the expected information gain for each of the $k$ candidate pairs. Each EIG evaluation requires computing the predictive label distribution and conditional posterior entropies across all particles, costing $\mathcal{O}(S)$ per pair. Thus, query selection under EIG costs $\mathcal{O}(kS)$ per iteration.

**Overall complexity.** Putting these components together, the dominant per-iteration cost is

$$\mathcal{O}(SD^2 + kS),$$

with occasional additional cost $\mathcal{O}(RSD^2)$ when rejuvenation is triggered. Over $T$ query rounds, the total runtime is therefore

$$\mathcal{O}\big(T(SD^2 + kS) + N_{\text{rej}} RSD^2\big),$$

where $N_{\text{rej}} \leq T$ is the number of resampling–rejuvenation events.

**Practical considerations.** In practice, $k \ll D^2$ and rejuvenation is triggered infrequently. Moreover, all particle-wise computations are embarrassingly parallel, making the method well suited to vectorized or GPU-accelerated implementations. For the problem sizes considered in our experiments ($D \leq 50$, $S \leq 1000$), CaPE runs comfortably within minutes on a

---

**Algorithm 4** Expert Response Model (synthetic oracle). The expert returns a label $y \in \{0, 1, 2\}$ according to the three-way likelihood $p_\theta(Y_{ij} = y \mid W^\star)$.

---

1: **function** QUERYEXPERT$(i, j, W^\star, \theta)$
2:    Compute the three-way probability vector

$$(p_0, p_1, p_2) = \big(p_\theta(Y_{ij} = 0 \mid W^\star),\, p_\theta(Y_{ij} = 1 \mid W^\star),\, p_\theta(Y_{ij} = 2 \mid W^\star)\big)$$

3:    Sample $y \in \{0, 1, 2\}$ from $\mathrm{Categorical}(p_0, p_1, p_2)$
4:    **return** $y$
5: **end function**

---

**Algorithm 5** Bayesian Weight Update and Resampling. Reweights particles using the expert likelihood, normalizes, and resamples if the effective sample size (ESS) drops below a threshold.

---

1: **function** BAYESIANUPDATE$(y, i, j, \{W^{(s)}, w^{(s)}\}, \theta)$
2:    **for** $s = 1$ **to** $S$ **do**
3:       $w^{(s)} \leftarrow w^{(s)} \, p_\theta(Y_{ij} = y \mid W^{(s)})$
4:    **end for**
5:    Normalize $w^{(s)} \leftarrow w^{(s)} / \sum_r w^{(r)}$
6:    **if** $\mathrm{ESS}(w) < \delta_s S$ **then**
7:       Resample $\{W^{(s)}\}$ proportional to $w^{(s)}$
8:       Reset $w^{(s)} = 1/S$
9:       REJUVENATEPARTICLES$(\{W^{(s)}\}, q_0, \mathcal{D}_t, \theta)$
10:   **end if**
11:   **return** $\{W^{(s)}, w^{(s)}\}$
12: **end function**

---

single CPU core, even under aggressive query budgets. For substantially larger graphs, the main bottleneck is evaluating or screening many candidate pairs. Practical extensions include restricting queries to candidate edges from an upstream discovery method, parallelizing particle-wise likelihood evaluations, or learning amortized proposal distributions for rejuvenation.

# F. Extended Related Work

The literature on causal discovery encompasses a wide range of algorithms, from classical constraint-based and score-based methods to more recent approaches that explicitly incorporate domain expertise and sequential querying. Below we provide an overview of the most relevant strands of research and position our work within this context.

## F.1. Classical Causal Discovery and Reliability Issues

Early work on causal inference formalized causal relationships through directed acyclic graphs (DAGs) and showed how DAGs encode conditional independence statements and can be interpreted causally under suitable assumptions (Pearl, 2009; Spirtes et al., 2001). Two broad algorithmic paradigms emerged: constraint-based methods that infer structure from conditional independence tests and score-based methods that search for DAGs that maximize a goodness-of-fit score. Constraint-based algorithms such as the PC algorithm and its variants use conditional independence tests to recover a Markov equivalence class and orient edges via deterministic rules. Score-based methods such as greedy equivalence search and Markov chain Monte Carlo (MCMC) sampling optimize a decomposable score (e.g., the Bayesian Information Criterion) over the space of DAGs. Despite their widespread use, both paradigms can be brittle when data are scarce. Modern approaches to DAG estimation have focused on exploiting recent breakthroughs in continuous characterizations of acyclicity (Zheng et al., 2018; Bello et al., 2022). However, these methods along with other recent frequentist approaches (Andrews et al., 2023) typically lack uncertainty quantification. In contrast, Bayesian methods do quantify uncertainty (see, e.g., Thompson et al., 2025; Annadani et al., 2023; Bonilla et al., 2026, for recent references) and can be used more naturally for expert knowledge elicitation in causal structure learning.

**Algorithm 6** Particle Rejuvenation. After resampling, each particle undergoes a small random mutation (add, delete, or flip an edge) accepted with a Metropolis–Hastings criterion under the current posterior. This step restores diversity and prevents particle collapse.

1: **function** REJUVENATEPARTICLES($\{W^{(s)}\}_{s=1}^S, q_0, \mathcal{D}_t, \theta$)
2: **for** $s = 1$ **to** $S$ **do**
3:    **if** particle selected for mutation **then**
4:       Propose a new graph $W'$ from $W^{(s)}$ by a small random edit:

- flip an existing edge orientation,

- add or remove a random edge,

- reject immediately if $W'$ is cyclic.

5:       Compute approximate log posterior difference

$$\Delta = \log q_0(W') - \log q_0(W^{(s)}) + \sum_{(i,j,y)\in\mathcal{D}_t} \big[ \log p_\theta(Y_{ij} = y \mid W') - \log p_\theta(Y_{ij} = y \mid W^{(s)}) \big].$$

6:       Accept $W'$ with probability $\min(1, e^\Delta)$; otherwise keep $W^{(s)}$.
7:    **end if**
8: **end for**
9: **return** updated particle set $\{W^{(s)}\}$
10: **end function**

## F.2. Expert Knowledge in Causal and Bayesian Network Structure Learning

Incorporating domain expertise into Bayesian network structure learning has a long history, most commonly by encoding knowledge *a priori* as structural constraints (e.g., required/forbidden edges) or as priors over structures and parameters that are combined with the data likelihood during score-based learning (Heckerman et al., 1995; Cooper & Herskovits, 1992). Constraint-based formulations that explicitly incorporate expert restrictions have also been studied, including exact and score-based learning under constraint sets (de Campos et al., 2009). In causal discovery, background knowledge is likewise used to restrict or orient parts of an equivalence class (e.g., by forbidding or requiring directions), refining partially directed graphs into more specific representations (Meek, 1995). Despite their practical importance, these approaches typically treat expert input as fixed and do not optimize which additional questions to ask when expert time is limited. Recent surveys emphasize that such priors and constraints are central to making structure learning usable in applied settings (Kitson et al., 2023).

Subsequent research developed more expressive forms of prior knowledge. Borboudakis & Tsamardinos (2012) introduced *path constraints*, which encode that a variable $X$ causally affects (or does not affect) another variable $Y$ through possibly indirect paths. Such knowledge may stem from experiments or temporal reasoning and can be formalized as a set $K$ of ordered pairs $(X, Y)$ labeled as causal ($X$ causes $Y$) or non-causal (no effect from $X$ to $Y$). The authors showed how to incorporate these constraints into partially directed acyclic graphs (PDAGs) and partial ancestral graphs (PAGs) by extending them to *path-constrained PAGs*, which force additional edge orientations and reduce structural uncertainties. They also demonstrated that even a few path constraints can lead to a large number of new inferences and summarized related methods that encode knowledge about parameters, direct relations, total orderings and complete structures. Complementary work by Chen et al. (2016) considered learning Bayesian networks under *ancestral constraints*, which specify that one variable is an ancestor or not of another. Because ancestral constraints are non-decomposable, most optimal structure learning algorithms cannot handle them directly; the authors proposed a framework that translates ancestral constraints into decomposable constraints for oracles and demonstrated orders-of-magnitude efficiency gains relative to integer-linear programming formulations.

More recently, variable grouping and typing constraints have been used to incorporate expert knowledge. For example, Parviainen & Kaski (2017) use expert-provided groups of variables to restrict the search space by enforcing that variables within a group share the same parents. Shojaie & Chen (2024) and Brouillard et al. (2022) introduce partial orders and

variable types to guide structure learning. These works show that incorporating structured prior information (whether through ancestral, group, order, or type constraints) can substantially reduce the number of admissible DAGs and improve recovery accuracy.

### F.3. Bayesian Experimental Design and Interventions

Bayesian experimental design formalizes the value of an experiment through the expected information it provides about unknown quantities of interest (Lindley, 1956; Chaloner & Verdinelli, 1995). In causal discovery, this principle is often used to select interventions that reduce uncertainty about causal structure (Tong & Koller, 2001; Murphy, 2001; Tigas et al., 2023; Toth et al., 2022; Annadani et al., 2024). In these settings, the action is an intervention $a$ and the observation is new data generated under $p(y \mid \mathrm{do}(a), W)$. CaPE uses the same information-theoretic principle in a different observation channel: the action is a structural query $(i, j)$ and the observation is a noisy categorical expert response $Y_{ij} \sim p_\theta(Y_{ij} \mid W)$. Thus, intervention BOED optimizes data acquisition, whereas CaPE optimizes knowledge elicitation. A direct empirical comparison would require a shared cost model for expert time and interventions, but the approaches are naturally complementary.

### F.4. Active and Expert-in-the-loop Causal Discovery

A parallel literature studies active causal discovery where the learner selects interventions or experiments to reduce uncertainty over causal structure. Early decision-theoretic approaches optimize expected information gain over Bayesian network structures when choosing experiments (Tong & Koller, 2001; Murphy, 2001), and more recent work has developed general frameworks for active Bayesian causal inference (Toth et al., 2022). These methods assume the learner can actively intervene on the system and observe new data. In contrast, our setting targets domains where interventions are fixed or unavailable, and instead leverages targeted human expert feedback as the primary source of additional information.

Recent work by Choo et al. (2023) formulates active causal structure learning with advice, where the learner is given side information about the DAG in the form of a purported graph. Their adaptive algorithm leverages this advice while still providing worst-case guarantees and builds on a long line of research on incorporating expert advice into causal discovery. They show that even imperfect advice can substantially reduce the number of interventions needed.

Human expertise can be incorporated not only through interventions but also through constraints and iterative feedback. The case study by Gururaghavendran & Murray (2024) demonstrates that although causal discovery algorithms can perform comparably to expert knowledge, novice use of these tools is not recommended: expert input is essential for selecting algorithms, choosing parameters, assessing assumptions and finalising adjustment sets. The study underscores the importance of integrating domain expertise into causal modelling and cautions against black-box use of causal discovery software.

Several recent works have explored interactive causal discovery, where users iteratively refine learned graphs using domain knowledge or are queried for information when uncertainty arises (Kitson & Constantinou, 2025). These approaches motivate selective elicitation of expert input and demonstrate benefits over fully static workflows. However, expert feedback is typically treated as deterministic constraints or heuristics, rather than as noisy observations integrated into a Bayesian posterior over causal graphs.

Expert-in-the-loop algorithms aim to refine causal models by systematically eliciting feedback from domain experts. Ankan & Textor (2025) propose a hybrid iterative approach that uses conditional independence testing to identify variable pairs where an edge is missing or superfluous and then consults an expert to add or remove edges. Their simulation study shows that if the expert orients edges correctly in at least two out of three cases, performance exceeds that of purely data-driven methods. In manufacturing, Ou et al. (2024) propose COKE, which leverages expert knowledge and chronological order information to learn causal graphs from data sets with up to 90% missing values. COKE maximizes the use of incomplete samples through an actor-critic optimisation procedure and achieves substantial F1-score improvements over benchmark methods.

Active GFlowNets (AGFNs), as proposed by da Silva et al. (2025), sample ancestral graphs (Richardson & Spirtes, 2002) with generative flow networks (GFNs, Bengio et al., 2021) guided by a score function, such as the Bayesian information criterion (BIC). These samples are then used to prompt an expert for feedback and update the model iteratively. AGFN and CaPE address the shared broad goal of interactive causal discovery, but they make different representational and algorithmic choices. AGFN operates on ancestral graphs, which can represent latent confounding through bidirected edges, and uses a four-category noisy relation variable together with a GFlowNet sampler. CaPE is not restricted to linear SEMs, operates on

DAG posteriors, uses a three-way local edge-existence/direction likelihood, and refines any set of posterior DAG particles through explicit Bayesian reweighting, resampling, and rejuvenation. The acquisition functions also differ: CaPE directly maximizes a BALD/mutual-information objective over the expert's categorical response, while AGFN ultimately uses a negative expected cross-entropy criterion for computational reasons. The practical tradeoff is therefore between the more general ancestral-graph representation of AGFN and the modular DAG-posterior refinement and tractable EIG acquisition of CaPE.

**Theoretical studies.** Ben-David & Sabato (2022) investigate the theoretical properties of active structure learning of Bayesian networks from observational data, when there are constraints on the number of variables that can be observed from the same sample. Related theoretical work studies the sample complexity of causal discovery without a conditional independence oracle and quantifies the value of domain expertise in terms of data samples (Wadhwa & Dong, 2021). In contrast, our theoretical results focus on uncertainty reduction, posterior contraction and identifiability within our proposed elicitation framework.

**Bayesian active learning and noisy supervision.** Our acquisition strategy is closely related to information-gain-based Bayesian active learning, notably BALD, which selects queries maximizing mutual information between model parameters and unobserved labels (Houlsby et al., 2011). Modeling experts as noisy annotators connects to classical work on estimating annotator error rates (Dawid & Skene, 1979). We bring these ideas into causal structure learning by defining a probabilistic expert response model over local edge relations and performing Bayesian updating over DAG posteriors under sequential queries.

### F.5. Summary

In summary, classical causal discovery methods recover Markov equivalence classes from data but are sensitive to statistical errors and faithfulness violations. Incorporating expert knowledge has been explored through priors on network structures, path or ancestral constraints, variable groupings and types, and active intervention design. Recent expert-in-the-loop algorithms highlight the value of sequentially eliciting expert feedback and updating posterior beliefs. Our work builds on these foundations by proposing a fully probabilistic framework that models uncertain expert feedback via a hierarchical logistic model, uses SMC to approximate and update the posterior over DAGs, and employs an information-gain acquisition function to ask the most informative questions. The main distinction is that CaPE treats expert input as a stochastic observation channel for posterior refinement rather than as a hard constraint or an intervention surrogate. By accommodating noisy responses and combining data and expert knowledge, our method aims to provide reliable causal discovery even in challenging settings.

## G. Experiments Details and Additional Results

### G.1. Static Baseline

As a strong non-adaptive baseline, we consider a static expected information gain policy (STE), which selects expert queries using the EIG criterion computed once from the initial observational posterior. Concretely, given an initial posterior $q_0(G \mid X)$ over DAGs and the expert response model $p_\theta(Y_{ij} \mid G)$, we compute for every ordered pair $(i, j)$ the expected information gain

$$\text{EIG}_0(i, j) = I_{q_0}(G ; Y_{ij}), \tag{19}$$

where the mutual information is taken with respect to the initial posterior $q_0$ and the expert's categorical response. The Static-EIG policy then ranks all candidate pairs by $\text{EIG}_0(i, j)$ and queries the top-ranked pairs sequentially, without updating the ranking as expert feedback is observed.

This baseline isolates the value of adaptivity in CaPE.

### G.2. Evaluation Metrics

This section summarizes the quantitative metrics used to evaluate the quality of posterior estimates and query policies in our synthetic experiments. Following recent discussion on Bayesian causal discovery evaluation (Karimi Mamaghan et al., 2024), we report posterior-sensitive quantities such as entropy, Brier score, and ETCP alongside graph summaries such as SHD, F1, AUPRC, and Top-$K$ precision. We distinguish between (i) probabilistic scoring rules applied to the posterior

predictive distribution, and (ii) structural metrics computed from posterior samples of the weighted DAG $W$.

### G.2.1. AVERAGE PREDICTIVE ENTROPY

This metric reflects uncertainty over local edge relations rather than graph-level uncertainty, making it sensitive to posterior concentration even when multiple DAGs remain globally plausible. Let $W$ denote a latent directed acyclic graph (DAG) and let $q_t(W)$ be the posterior distribution over DAGs after $t$ rounds of expert feedback, represented by a weighted particle approximation. For any ordered variable pair $(i, j)$, the model induces a posterior predictive distribution over expert responses $Y_{ij} \in \{0, 1, 2\}$, corresponding to the judgements $i \rightarrow j$, $j \rightarrow i$, or no direct edge. This distribution is obtained by marginalizing the expert response likelihood over the posterior,

$$p_t(Y_{ij} = y) \;=\; \mathbb{E}_{W \sim q_t}[p_\theta(Y_{ij} = y \mid W)].$$

The predictive entropy for pair $(i, j)$ at round $t$ is defined as

$$H_t(i, j) \;=\; -\sum_{y \in \{0,1,2\}} p_t(Y_{ij} = y) \log p_t(Y_{ij} = y).$$

To summarize global posterior uncertainty, we report the *average predictive entropy* over the candidate set of queried or queryable pairs $\mathcal{C}_t$ considered at round $t$,

$$\bar{H}_t \;=\; \frac{1}{|\mathcal{C}_t|} \sum_{(i,j) \in \mathcal{C}_t} H_t(i, j).$$

As expert feedback is incorporated and the posterior concentrates, the predictive distributions $p_t(Y_{ij})$ become increasingly peaked, leading to a decrease in $\bar{H}_t$.

### G.2.2. PROBABILISTIC SCORING METRICS

For any ordered pair $(i, j)$, the posterior predictive distribution over expert responses is

$$\widehat{p}_t^{ij}(y) = \sum_{s=1}^{S} w_t^{(s)} p_\theta(Y_{ij} = y \mid W^{(s)}), \qquad y \in \{0, 1, 2\}.$$

Let $y_{ij}^\star$ denote the ground-truth label implied by the true DAG $W^\star$. We define two proper scoring rules.

**Expected true-class probability (ETCP).** This measures how much probability mass the posterior predictive model assigns to the true response:

$$\mathrm{ETCP}_t = \frac{1}{D(D-1)} \sum_{\substack{i,j=1 \\ i \neq j}}^{D} \widehat{p}_t^{ij}(y_{ij}^\star).$$

Larger values indicate better calibrated predictive beliefs.

**Brier score.** The multiclass Brier score averages squared prediction error over the three possible labels:

$$\mathrm{Brier}_t = \frac{1}{D(D-1)} \sum_{\substack{i,j=1 \\ i \neq j}}^{D} \sum_{y \in \{0,1,2\}} \left(\widehat{p}_t^{ij}(y) - \mathbf{1}\{y = y_{ij}^\star\}\right)^2.$$

Smaller values indicate sharper and better calibrated predictions.

### G.3. Posterior-marginal Edge Summaries

For each ordered pair $(i, j)$, the posterior probability of a directed edge is

$$\pi_{ij}^{\mathrm{exist}}(t) = \sum_{s=1}^{S} w_t^{(s)} \mathbf{1}\left[W_{ij}^{(s)} \neq 0\right].$$

We also define the posterior orientation probability conditional on an edge existing:

$$\pi_{ij}^{\text{orient}}(t) = \frac{\sum_s w_t^{(s)} \mathbf{1}[W_{ij}^{(s)} \neq 0]}{\sum_s w_t^{(s)} \mathbf{1}[W_{ij}^{(s)} \neq 0 \ \vee \ W_{ji}^{(s)} \neq 0]}.$$

These marginal quantities provide uncertainty-aware summaries but do not directly evaluate structural accuracy; for that we rely on sample-based metrics below.

### G.3.1. SAMPLE-BASED STRUCTURAL METRICS

For each particle $W^{(s)}$, let $A^{(s)}$ denote its binary adjacency matrix and let $A^\star$ denote the adjacency matrix of the true DAG $W^\star$. Define

$$\widehat{A}_{ij}^{(s)} = \mathbf{1}[W_{ij}^{(s)} \neq 0], \qquad A_{ij}^\star = \mathbf{1}[W_{ij}^\star \neq 0].$$

Structural metrics are computed per particle and then averaged over the posterior.

**Skeleton F1.** Let

$$\text{Skel}(A) = \{\{i, j\} : A_{ij} = 1 \text{ or } A_{ji} = 1\}$$

denote the undirected skeleton of a DAG. For particle $s$, define precision and recall as

$$\text{Prec}_{\text{skel}}^{(s)} = \frac{|\text{Skel}(A^{(s)}) \cap \text{Skel}(A^\star)|}{|\text{Skel}(A^{(s)})|}, \qquad \text{Rec}_{\text{skel}}^{(s)} = \frac{|\text{Skel}(A^{(s)}) \cap \text{Skel}(A^\star)|}{|\text{Skel}(A^\star)|}.$$

The skeleton F1 score is

$$\text{F1}_{\text{skel}}^{(s)} = \frac{2\,\text{Prec}_{\text{skel}}^{(s)} \text{Rec}_{\text{skel}}^{(s)}}{\text{Prec}_{\text{skel}}^{(s)} + \text{Rec}_{\text{skel}}^{(s)}}.$$

The reported score averages $\text{F1}_{\text{skel}}^{(s)}$ over $s$.

**Orientation F1.** Orientation evaluation considers only edges present in the skeleton of either graph. Let

$$\text{Orient}(A) = \{(i, j) : A_{ij} = 1\}$$

denote the set of directed edges. Define precision and recall:

$$\text{Prec}_{\text{orient}}^{(s)} = \frac{|\text{Orient}(A^{(s)}) \cap \text{Orient}(A^\star)|}{|\text{Orient}(A^{(s)})|}, \qquad \text{Rec}_{\text{orient}}^{(s)} = \frac{|\text{Orient}(A^{(s)}) \cap \text{Orient}(A^\star)|}{|\text{Orient}(A^\star)|}.$$

The particle-level orientation F1 score is

$$\text{F1}_{\text{orient}}^{(s)} = \frac{2\,\text{Prec}_{\text{orient}}^{(s)} \text{Rec}_{\text{orient}}^{(s)}}{\text{Prec}_{\text{orient}}^{(s)} + \text{Rec}_{\text{orient}}^{(s)}},$$

and it is averaged over posterior samples.

**Structural Hamming distance (SHD).** The SHD between two DAGs counts the minimum number of edge insertions, deletions, or orientation flips required to make them identical. For particle $s$,

$$\text{SHD}^{(s)} = \sum_{i \neq j} \mathbf{1}\left[A_{ij}^{(s)} \neq A_{ij}^\star\right].$$

Orientation differences count as one mismatch. We report the posterior-averaged SHD

$$\text{SHD}_t = \sum_{s=1}^{S} w_t^{(s)} \text{SHD}^{(s)}.$$

**Summary.** Probabilistic scoring rules evaluate the calibration and sharpness of the posterior predictive distribution over expert responses $Y_{ij}$, while structural metrics measure the accuracy of the recovered DAG relative to the ground truth. Taken together, these metrics quantify both local predictive quality and global structural recovery of the causal graph.

## G.4. Synthetic Data Settings

We evaluate our framework on controlled synthetic causal discovery tasks in which both observational data and expert responses are generated from a known ground-truth directed acyclic graph (DAG). The experiment is designed to test whether expert feedback, when combined with Bayesian active learning, drives posterior concentration toward the true graph.

**Graph generation.** For each trial we sample a ground-truth weighted DAG on $D = 20$ nodes using the Erdős–Rényi model with edge probability $p_{\text{true}} = 0.25$. Nonzero edge weights are drawn independently from $\text{Uniform}[0.5, 1.5]$. This produces moderately sparse graphs with an expected number of approximately $D(D-1)p_{\text{true}}/2 = 47.5$ directed edges under the sampled topological order.

**Prior distribution.** To emulate a noisy or imperfect observational posterior, we construct an initial particle approximation $q_0(W \mid X)$ by perturbing the ground-truth DAG with three forms of random noise. This controlled construction is used only in the synthetic experiment to isolate the behavior of the acquisition policy under known posterior uncertainty. It is not required by CaPE: the Sachs, CausalBench, and DAG-GFN experiments use data-driven posterior samples, and the framework can in principle start from any prior over DAGs.

- edge flips with probability 0.10,

- random edge additions or removals with probability 0.05,

- independent additive noise on nonzero weights with standard deviation 0.20.

We draw $S = 10{,}000$ particles from this perturbed prior and use them as the initial posterior representation.

**Expert model.** When queried on a pair $(i, j)$, the expert returns a label $Y_{ij} \in \{0, 1, 2\}$ corresponding to $\{j \to i, \ i \to j, \ \text{none}\}$. Responses are drawn from the hierarchical three-way logistic model described in Section 4, with parameters

$$\beta_{\text{edge}} = 10.0, \qquad \beta_{\text{dir}} = 10.0, \qquad \gamma = 0.1, \qquad \lambda = 0.$$

This yields a highly informative expert.

**Query policy.** Each run consists of $T = 190$ query rounds. The long trajectory is used to characterize posterior-concentration dynamics, not to imply that such a budget is necessary in deployment; the real-data results also report meaningful gains at small budgets. Particle rejuvenation with two Metropolis–Hastings steps is applied whenever the effective sample size falls below $0.6S$.

**Evaluation.** At each round we compute all metrics, where posterior marginals are obtained by averaging over the weighted particle set. Results are aggregated over 10 random seeds.

## G.5. Details of the Sachs Dataset and Settings

### G.5.1. DESCRIPTION

We evaluate our method on the protein signaling dataset introduced by Sachs et al. (2005), which has become a standard benchmark for causal discovery. The dataset consists of multiparameter flow-cytometry measurements collected from human T cells, recording the activity levels of key proteins involved in intracellular signaling pathways. Specifically, the data contain continuous measurements of 11 signaling proteins (*Raf*, *Mek*, *Plcg*, *PIP2*, *PIP3*, *Erk*, *Akt*, *PKA*, *PKC*, *P38*, and *Jnk*) under a range of experimental conditions.

The original experiments include both observational samples and samples collected under targeted chemical interventions that perturb specific components of the signaling network. Following common practice in the causal discovery literature, we

restrict attention to the observational subset of the data and evaluate learned structures against the reference causal graph reported by Sachs et al. (2005). This reference graph contains 11 nodes and 17 directed edges and is widely used as a ground-truth proxy in benchmarking studies.

We obtain the observational data from the `bnlearn` repository (https://www.bnlearn.com/book-crc/code/sachs.data.txt.gz), which provides a curated and standardized version of the dataset consistent with the original publication. All variables are treated as continuous. As is standard for this benchmark, we assume causal sufficiency and model the system using directed acyclic graphs, despite the presence of feedback loops in biological signaling, to enable comparison with prior work.

### G.5.2. SETTINGS

We construct an initial observational posterior $q_0(G \mid X)$ using a bootstrap-based linear DAG sampler. Each particle is generated by sampling a random variable ordering, selecting candidate parents via correlation screening, and fitting ridge-regularized linear regressions. This yields a diverse set of data-consistent DAGs without imposing hard structural constraints. See Section H for more details.

The posterior is represented using $S = 500$ particles and updated sequentially as expert feedback is acquired. At each round, we query an expert about the existence and orientation of a single edge, modeled as a three-way categorical response. Queries are selected adaptively by maximizing expected information gain (EIG) under the current posterior, after screening to the most uncertain candidate pairs. We run the interaction for a total of $T = 40$ expert queries per run.

Posterior inference is performed via importance reweighting with effective-sample-size-based resampling, and we apply a lightweight Metropolis–Hastings rejuvenation step to maintain particle diversity. All results are averaged over 10 random seeds. Hyperparameters are in Table 1.

*Table 1.* Sachs experiment hyperparameters.

| Component | Setting |
|---|---|
| Particles ($S$) | 500 |
| Query budget ($T$) | 40 |
| Query policy | Adaptive EIG |
| Screened pairs ($k$) | 200 |
| Expert likelihood ($\beta_{\text{edge}}, \beta_{\text{dir}}, \lambda$) | (10.0, 10.0, 0.0) |
| ESS resample threshold | 0.5 |
| Rejuvenation | Enabled (2 MH steps) |
| Max parents per node | 3 |
| Correlation screening $k$ | 6 |
| Ridge coefficient | $10^{-3}$ |

### G.6. Details of the CausalBench Dataset and Settings

#### G.6.1. DESCRIPTION

**Dataset provenance and structure.** We use the CRISPR perturbation dataset distributed as part of CausalBench (Chevalley et al., 2025, https://github.com/causalbench). The data consist of gene expression measurements in the K562 human leukemia cell line collected after targeted single-gene perturbations. Each interventional sample is annotated with metadata indicating the Ensembl gene identifier (prefix ENSG) of the perturbed gene following the Ensembl genome annotation system (Yates et al., 2019), providing a natural source of localized interventional information; additional samples correspond to non-targeting controls. Importantly, these annotations specify the experimental intervention and do not constitute supervised labels inferred from expression.

**Problem formulation.** We construct a 50-node problem instance by selecting the $K = 50$ genes with the highest marginal variance in the observational subset of the data, a standard preprocessing step in transcriptomic analysis that yields a tractable yet nontrivial causal inference task. The resulting observational data matrix has $N = 8,553$ observations and $D = 50$ variables.

**Construction of the oracle effect graph from perturbations.** Unlike classical benchmark settings with a DAG ground truth, gene perturbation data naturally induces an *effect graph* that can contain cycles and feedback. We therefore define the oracle to provide local causal feedback using the interventional samples. Following Chevalley et al. (2025), we define a directed binary *effect graph* that summarizes the downstream transcriptional consequences of perturbations. Specifically, for each perturbed gene $i$ and each measured gene $j$, we test whether perturbing $i$ induces a practically meaningful distributional change in $X_j$ relative to non-targeting controls. Concretely, we compare the perturbed and control marginals using a two-sample Kolmogorov–Smirnov test, apply Benjamini–Hochberg false discovery rate control at level $\alpha = 0.05$, and additionally require an effect-size threshold on the absolute mean shift (we use min_effect = 0.3). We set an inclusion threshold of min_group_n = 25 samples per perturbation; in this dataset all selected perturbations meet this criterion. This procedure yields a directed binary adjacency matrix $A^\star \in \{0,1\}^{K \times K}$, where $A^\star_{ij} = 1$ indicates that perturbing gene $i$ has a significant and practically meaningful effect on gene $j$. On the resulting 50-node instance, the oracle effect graph has density $127/(K(K-1)) \approx 0.0518$. Because perturbation effects may involve feedback and indirect regulation, $A^\star$ is not constrained to be acyclic and may contain cycles.

**Handling of bidirectional effects.** Our expert query interface returns three-way categorical responses and therefore cannot represent bidirectional effects on a single query. To ensure a well-defined oracle, we conservatively remove pairs $(i, j)$ for which both $A^\star_{ij} = 1$ and $A^\star_{ji} = 1$ when generating oracle responses. These pairs are treated as ambiguous and mapped to the "no directed effect" category. This choice avoids introducing spurious directionality while preserving the majority of directed signal in the oracle graph.

G.6.2. Settings

We construct the initial observational posterior $q_0(G \mid X)$ from the observational component of the CausalBench Weissmann K562 dataset. As in the Sachs experiments, the posterior is represented by a particle approximation, where each particle corresponds to a candidate linear-Gaussian DAG fitted to bootstrap resamples of the observational data. Parent sets are restricted via correlation-based screening, and edge weights are estimated using ridge-regularized linear regression. This procedure yields a diverse ensemble of data-consistent DAGs while preserving substantial structural uncertainty. See Section H for details.

To obtain a tractable benchmark, we restrict attention to a 50-node subgraph selected by ranking genes according to marginal variance in the observational data. A perturbation-derived effect graph, constructed from interventional data, is used exclusively as an oracle to simulate expert feedback and to compute evaluation metrics.

The posterior is represented using $S = 1000$ particles and updated sequentially as expert feedback is acquired. At each round, the learner queries an expert about the existence and orientation of a single unordered node pair, modeled as a three-way categorical response. Queries are selected adaptively by maximizing expected information gain (EIG) under the current posterior, after screening to the most uncertain candidate pairs. Each run proceeds for a total of $T = 200$ expert queries.

Posterior inference is performed via importance reweighting with effective-sample-size-based resampling, and a lightweight Metropolis–Hastings rejuvenation step is applied to maintain particle diversity. All results are averaged over 10 random seeds. Hyperparameters are reported in Table 2.

**Evaluation protocol.** Because the oracle defines an effect graph rather than a ground-truth DAG, classical structural metrics such as SHD are not appropriate. Instead, we treat posterior edge marginals as directed edge scores and evaluate ranking performance against $A^\star$ using directed AUPRC and AUROC, as well as Top-$K$ precision, where $K$ is set to the number of positive oracle edges. This evaluation protocol aligns with prior work on perturbation-based causal benchmarks and emphasizes the recovery of experimentally supported directional effects rather than exact graph topology (Chevalley et al., 2025; Peters et al., 2017).

# H. Bootstrap-Based Observational Prior

To initialize the posterior over causal graphs, we construct an observational prior $q_0(G \mid X)$ using a bootstrap-based linear DAG sampling procedure. The goal of this procedure is to generate a diverse collection of DAGs that are consistent with the observed data while remaining computationally lightweight and agnostic to the downstream expert elicitation process.

*Table 2.* CausalBench-50 experiment hyperparameters.

| Component | Setting |
|---|---|
| Particles ($S$) | 1000 |
| Query budget ($T$) | 200 |
| Query policy | Adaptive EIG |
| Screened pairs ($k$) | 800 |
| Expert likelihood ($\beta_{\text{edge}}, \beta_{\text{dir}}, \lambda$) | (10.0, 10.0, 0.0) |
| ESS resample threshold | 0.5 |
| Rejuvenation | Enabled (2 MH steps) |
| Max parents per node | 3 |
| Correlation screening $k$ | 8 |
| Ridge coefficient | $10^{-2}$ |
| Bootstrap sample size | Full $N$ (no subsampling) |

Given an observational data matrix $X \in \mathbb{R}^{N \times D}$, we generate a set of $S$ DAG particles as follows. For each particle, we first draw a bootstrap resample of the data by sampling $N$ rows of $X$ with replacement. We then sample a random permutation of the $D$ variables, which defines a total ordering that guarantees acyclicity.

Variables are processed sequentially according to this order. When considering a variable $X_j$, we restrict candidate parents to variables that appear earlier in the order. Among these candidates, we compute marginal correlations with $X_j$ and retain only the top $k$ variables with the largest absolute correlation. From this screened set, we further restrict the number of parents to at most $p_{\max}$ variables. The conditional relationship between $X_j$ and its selected parents is then modeled using ridge-regularized linear regression, and directed edges are added for regression coefficients whose magnitude exceeds a small threshold.

Repeating this procedure yields a collection of weighted adjacency matrices, each representing a DAG consistent with a particular bootstrap sample and variable ordering. The resulting set of graphs forms a particle approximation to the initial observational posterior $q_0(G \mid X)$, which is subsequently refined using expert feedback. This construction intentionally avoids hard structural constraints and produces a broad prior that reflects uncertainty due to finite data and model variability. *We emphasize that this procedure is used solely to initialize the posterior and does not constitute a causal discovery algorithm on its own.*

### H.1. Details of the DAG-GFN Prior on Sachs

We train a trajectory-balance GFlowNet to sample directed acyclic graphs with probability proportional to $\exp(\text{BIC}(G; X)/\tau_g)$, where $\text{BIC}(G; X)$ denotes the linear-Gaussian Bayesian Information Criterion computed on the Sachs observational dataset and $\tau_g$ is a temperature parameter. The resulting GFlowNet defines a data-driven prior distribution over DAGs that captures high-scoring observational structures without incorporating expert feedback.

### H.2. Misspecification Robustness Details

We report additional synthetic experiments in which the oracle expert deviates from the likelihood assumed by CaPE. In all regimes, inference keeps the same model and hyperparameters as the main paper, $(\beta_{\text{edge}}, \beta_{\text{dir}}) = (10, 10)$; only the response-generating oracle changes. The regimes are summarized as follows:

- **Pairwise heterogeneous reliability:** reliability varies across queried pairs, $(\beta^\star_{\text{edge},ij}, \beta^\star_{\text{dir},ij}) \sim \text{Uniform}[2, 18]$, independently.

- **Systematic directional bias:** responses are first generated by the matched logistic oracle and then directional answers are flipped with probability $p_{\text{flip}} = 0.15$.

- **Abstention bias:** responses are first generated by the matched logistic oracle and then directional answers are converted to the "none" category with probability $p_{\text{none}} = 0.20$.

- **Very noisy expert:** $(\beta^\star_{\text{edge}}, \beta^\star_{\text{dir}}) = (1.0, 1.0)$.

- **Extremely noisy expert:** $(\beta^\star_{\text{edge}}, \beta^\star_{\text{dir}}) = (0.5, 0.5)$.

These regimes test qualitatively different non-adversarial deviations from the assumed expert model: pair-dependent reliability, structured directional corruption, abstention behavior, and substantially lower informativeness.

The results in Figure 7 show that CaPE remains competitive across all regimes. The EIG policy is most beneficial when the posterior contains structural disagreements that map to informative expert responses; as the expert becomes extremely noisy, all methods improve more slowly, but CaPE does not collapse under the mismatched likelihood.

### H.3. Additional Results

#### H.3.1. SYNTHETIC DATA

Additional results in Figure 8.

#### H.3.2. SACHS

Additional results in Figure 9 and Table 3.

| Metric | Policy | $T = 5$ | $T = 10$ | $T = 20$ | $T = 40$ |
|---|---|---|---|---|---|
| SHD $\downarrow$ | EIG | $25.56 \pm 2.09$ | $23.34 \pm 1.81$ | $17.76 \pm 1.75$ | $15.09 \pm 2.79$ |
| SHD $\downarrow$ | Uncertainty | $25.70 \pm 1.40$ | $23.87 \pm 1.48$ | $20.03 \pm 2.65$ | $16.45 \pm 2.95$ |
| SHD $\downarrow$ | Random | $25.77 \pm 1.52$ | $23.76 \pm 1.57$ | $21.53 \pm 1.63$ | $17.29 \pm 3.32$ |
| Orient. F1 $\uparrow$ | EIG | $0.300 \pm 0.062$ | $0.357 \pm 0.051$ | $0.495 \pm 0.052$ | $0.600 \pm 0.070$ |
| Orient. F1 $\uparrow$ | Uncertainty | $0.311 \pm 0.034$ | $0.368 \pm 0.046$ | $0.453 \pm 0.066$ | $0.555 \pm 0.062$ |
| Orient. F1 $\uparrow$ | Random | $0.307 \pm 0.048$ | $0.363 \pm 0.059$ | $0.423 \pm 0.063$ | $0.545 \pm 0.086$ |
| $\Delta$SHD $\downarrow$ | EIG | $-3.22 \pm 2.04$ | $-5.44 \pm 1.75$ | $-11.02 \pm 1.75$ | $-13.69 \pm 2.79$ |
| $\Delta$SHD $\downarrow$ | Uncertainty | $-3.07 \pm 1.41$ | $-4.91 \pm 1.46$ | $-8.75 \pm 2.59$ | $-12.33 \pm 2.92$ |
| $\Delta$SHD $\downarrow$ | Random | $-3.00 \pm 1.53$ | $-5.02 \pm 1.60$ | $-7.24 \pm 1.62$ | $-11.49 \pm 3.32$ |
| $\Delta$Orient. F1 $\uparrow$ | EIG | $0.082 \pm 0.062$ | $0.138 \pm 0.050$ | $0.277 \pm 0.050$ | $0.382 \pm 0.069$ |
| $\Delta$Orient. F1 $\uparrow$ | Uncertainty | $0.093 \pm 0.034$ | $0.150 \pm 0.046$ | $0.234 \pm 0.065$ | $0.336 \pm 0.061$ |
| $\Delta$Orient. F1 $\uparrow$ | Random | $0.088 \pm 0.048$ | $0.145 \pm 0.059$ | $0.204 \pm 0.062$ | $0.327 \pm 0.085$ |

*Table 3.* Sachs observational-only benchmark: mean $\pm$ std across runs at different expert-query budgets.

#### H.3.3. CAUSALBENCH

Additional results in Figure 10.

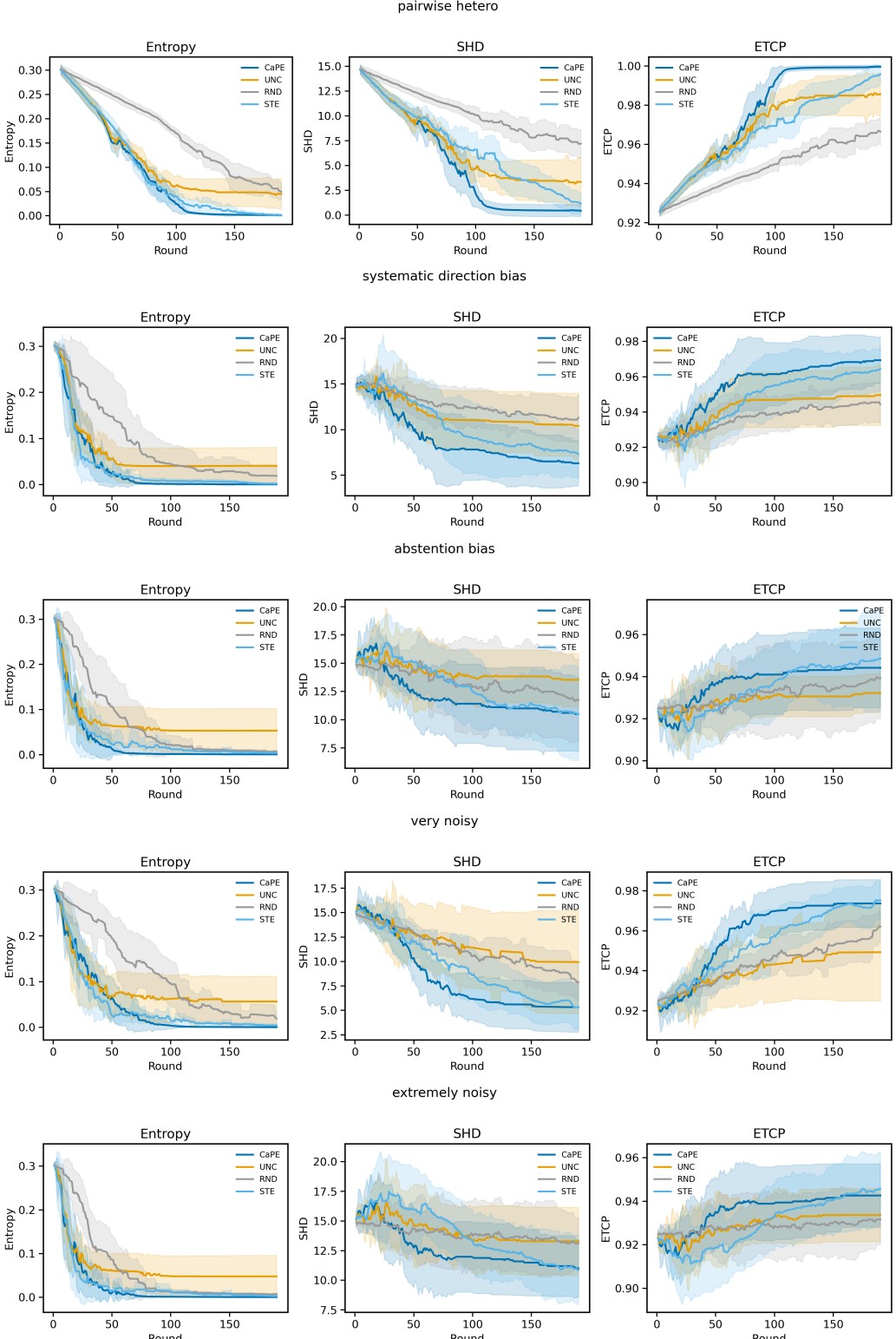

*Figure 7.* Full robustness trajectories under expert-model misspecification: entropy (↓), SHD (↓), and ETCP (expected true class probability, ↑). From top to bottom: pairwise heterogeneous reliability, systematic directional bias, abstention bias, very noisy experts, and extremely noisy experts.

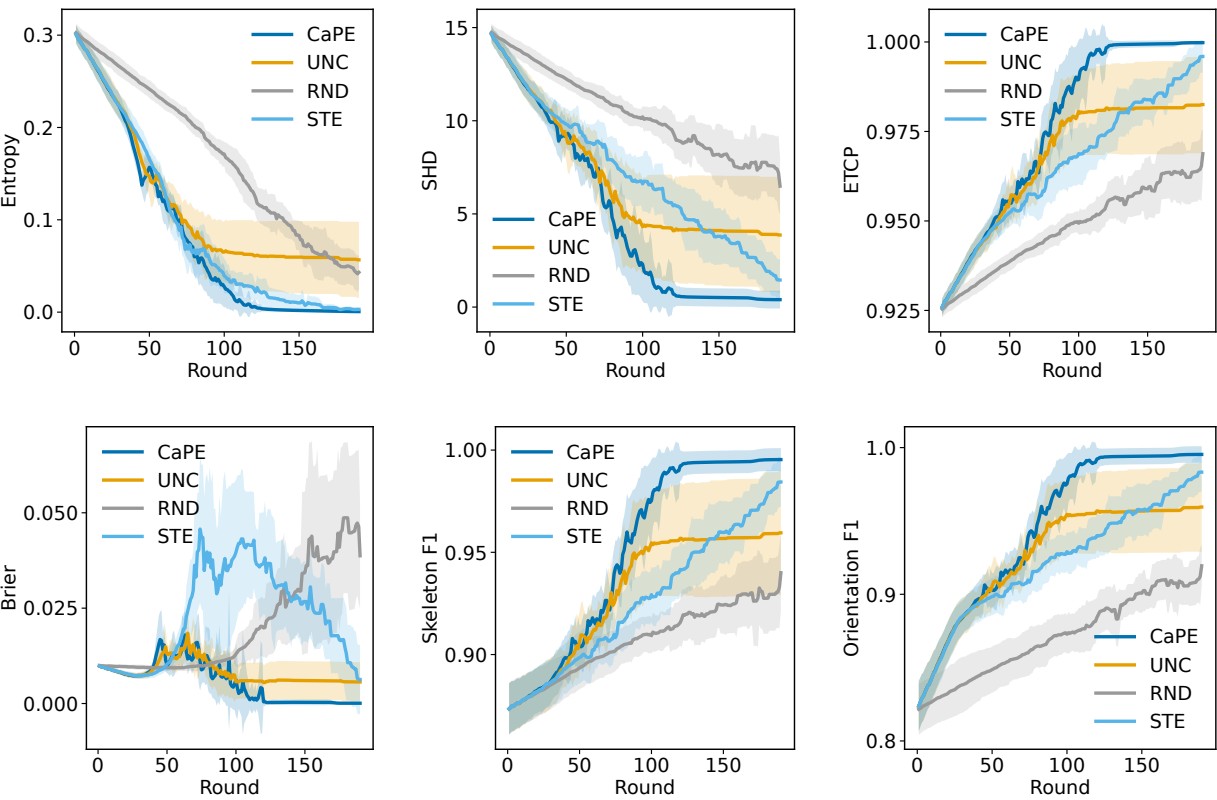

*Figure 8.* Full results on synthetic data: entropy (↓), SHD (↓), ETCP (expected true class probability, ↑) , Brier (↓), Skeleton F1 (↑), and Orientation F1 (↑).

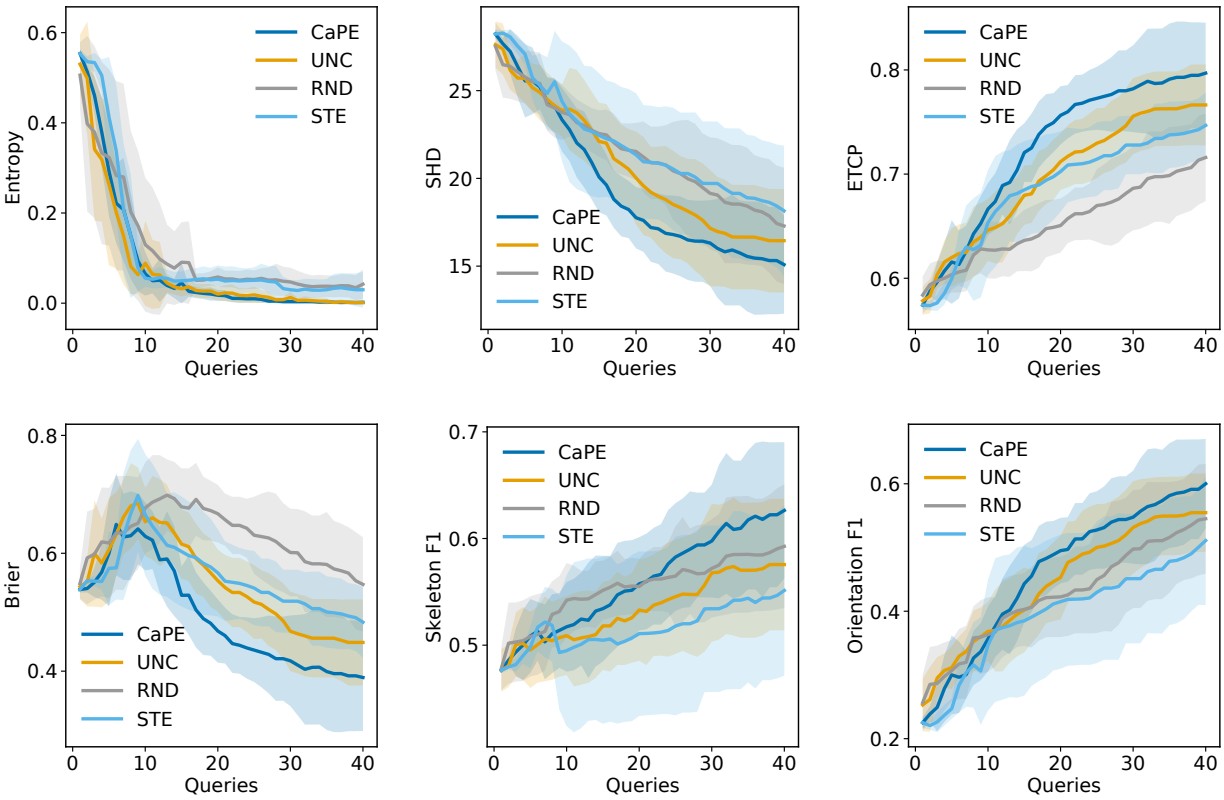

*Figure 9.* Full results on Sachs data: entropy (↓), SHD (↓), ETCP (expected true class probability, ↑) , Brier (↓), Skeleton F1 (↑), and Orientation F1 (↑).

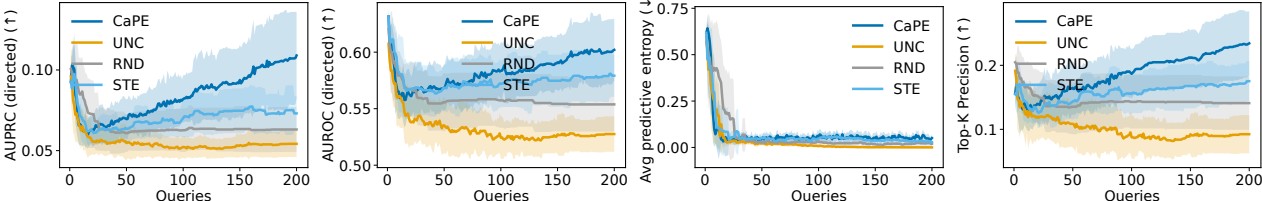

*Figure 10.* Full results on Causalbench.

