# OpenReview forum: "Causal Preference Elicitation"
_ICML.cc/2026/Conference — ICML 2026 regular_

### Official Review · Reviewer_WupM · 2026-02-27

**Soundness:** 3
**Presentation:** 3
**Significance:** 3
**Originality:** 3
**Overall Recommendation:** 4
**Confidence:** 2

**Summary:**

This paper proposes a causal preference elicitation framework that combines the modification of the likelihood model, particle-based approximate posterior for inference, and the expected information gain (EIG) function for the data acquisition. Later, authors show that 1. maximizing EIG is equivalent to minimizing the KL-divergence between the current posterior and the posterior after observing the expert's answer, and 2. Their framework reaches convergence to the true causal graph. Finally, the authors test their framework on synthetic and real-world datasets.

**Compliance With Llm Reviewing Policy:**

Affirmed.

**Final Justification:**

I am satisfied with the author's response. However, I have lower confidence in this field. Therefore, I will maintain my initial score.

**Key Questions For Authors:**

- What is the reason authors use the AUPRC metric instead of the AUROC?
- Is it possible to apply this framework without running the standard causal discovery first?

**Limitations:**

- Authors should mention the limitations of their data acquisition and the chosen approximate inference technique.
- The requirements of running standard causal discovery first can also be considered as limitations.

**Strengths And Weaknesses:**

Soundness:
- The likelihood model sounds reasonable, as it considers the complete possibility of the DAG (existences + directions).
- This paper assumes we have to run the standard causal discovery method first before involving the preference feedback from humans. This fact makes the proposed framework less efficient, as I expect the framework can directly involve a human expert from the start.
- Authors do not provide conceptual and/or intuitive justification for the choice of each component of their framework, e.g., particle-based approximation for the inference, and EIG for the data acquisition.
- The particle-based approximation requires further preprocessing steps (particle rejuvenation and resampling based on ESS results) to work well. It seems like this approximation technique is not naturally suitable for the problem. I am wondering why authors did not consider other approximation techniques.
- EIG is a common and principal data acquisition for active learning, and it seems fit to the problem addressed by this paper.
- Before optimizing the acquisition function, this paper proposes to sort the top-k candidates with another measure. It looks like the acquisition phase becomes a greedy optimization, which may result in a suboptimal query. Maybe authors can clarify this.

Presentation:
- The flow of the manuscript is nice. Each section orderly explains the component of the framework.
- The notation seems standard and understandable.

Significance:
- This work can be one of the references for solving the causal discovery problem that involves human experts. The method looks practical, and I expect this method can be applied in many fields.

Originality:
This paper leverages existing methods to build a framework that seems to work harmoniously with itself after some tweaks.

---

> ### Author Rebuttal · Authors · 2026-03-30
>
> We thank the reviewer for the positive assessment and helpful suggestions. We address the main questions below.
>
> **1. Choice of particle-based inference and EIG**
>
> Our goal is to maintain and update a posterior over DAGs under sequential, noisy expert feedback. Particle-based inference provides a flexible way to approximate this posterior while supporting Bayesian updates of the form $q\_t(W) \propto q\_{t-1}(W) p(y\_t\mid W)$ after each query. While this requires standard SMC steps (resampling/rejuvenation), it allows us to represent multi-modal uncertainty over graph structures, which is important in causal discovery.
>
> EIG is used as it directly measures the expected reduction in posterior uncertainty. Intuitively, it prioritizes queries for which candidate DAGs in the current posterior disagree most, making it well-suited for resolving structural ambiguities.
>
> **2. Candidate screening and acquisition**
>
> The top-$k$ screening step is used for computational efficiency, as evaluating EIG over all $O(D^2)$ pairs at every round is expensive. Importantly, the final selection is still based on EIG, so the procedure remains guided by the same information-theoretic objective. We will clarify this point.
>
> **3. Use of initial causal discovery**
>
> We initialize from an observational posterior to capture data-driven structural uncertainty before incorporating expert feedback. This improves efficiency by focusing queries on unresolved edges. However, the framework itself does not require a specific causal discovery method and can, in principle, incorporate expert feedback from the start given any initial prior (including uniform and non-informative priors). We will clarify this flexibility.
>
> **4. Evaluation metric (AUPRC vs AUROC)**
>
> We use AUPRC because the adjacency matrix is typically sparse, making precision-recall more informative than AUROC in this setting. We will clarify this choice in the paper.
>
> **5. Limitations**
>
> We agree that discussing limitations would strengthen the paper. In particular, we will add a dedicated section covering assumptions in the expert model, the computational cost of particle-based inference, and the reliance on an initial observational posterior.

---

> > ### Author Rebuttal · Reviewer_WupM · 2026-04-02
> >
> > All of my concerns have been resolved, and I maintain my initial score.

---

> > > ### Author Response · Authors · 2026-04-04
> > >
> > > We are glad that we were able to address your concerns. We would appreciate if you would consider raising your score accordingly, or to let us know of any remaining concerns you have that keep you from doing so.

---

### Official Review · Reviewer_HoX7 · 2026-03-10

**Soundness:** 3
**Presentation:** 3
**Significance:** 2
**Originality:** 3
**Overall Recommendation:** 4
**Confidence:** 4

**Summary:**

This paper presents Causal Preference Elicitation (CaPE), a sequential approach to Bayesian Causal Discovery (BCD) in which an expert provides feedback on whether an edge exists (and its direction) between a queried pair of nodes in an unknown causal graph. The authors assume access to an initial observational posterior distribution, which can be obtained from any existing BCD method. This posterior is then refined iteratively in a sequential manner based on expert feedback.

The design space consists of selecting a specific entry in the adjacency matrix to query, and this selection is performed probabilistically using a Bayesian Optimal Experimental Design (BOED) framework. Specifically, this is equivalent to maximizing expected information gain under the BALD objective. The method is evaluated on both synthetic and real-world datasets, offering an alternative approach to causal discovery that integrates expert feedback with Bayesian sequential structure learning.

**Compliance With Llm Reviewing Policy:**

Affirmed.

**Key Questions For Authors:**

1. If the algorithm repeatedly selects the same pair of variables for querying due to posterior uncertainty, then under the assumed likelihood model (with i.i.d. expert responses), one would need to consult different independent experts each time. How feasible is this in practice? Moreover, if an expert is uncertain about the existence of an edge, does a sequential querying process meaningfully improve the situation?

2. Relatedly, if an expert is unsure whether an edge exists, performing an intervention may be a more reliable way to obtain informative data. Would it be beneficial to combine expert feedback with an intervention design framework? There has been substantial work on amortized probabilistic intervention design (e.g., Annadani et al., 2024), which could potentially be integrated with CaPE to yield a more holistic and practically deployable framework. I would be interested in the authors’ perspective on this direction.

3. Have the authors considered evaluating additional metrics commonly used in the BCD community? Metrics such as SHD and AUPRC may not fully reflect the quality of the approximated posterior distribution (see discussion in Karimi Mamaghan et al., 2024). Alternative posterior-sensitive evaluations could provide deeper insight.

**Limitations:**

There is no separate limitations section.Explicitly discussing the practical challenges and assumptions outlined above would improve the completeness of the paper.

### References

Lindley, On a measure of the information provided by an experiment. Annals of Mathematical Statistics (1956).

Chaloner & Verdinelli, Bayesian experimental design: A review. Statistical Science (1995).

Tigas et al.,  Differentiable Multi-Target Causal Bayesian Experimental Design (ICML 2023).

Karimi Mamaghan et al., Challenges and considerations in the evaluation of Bayesian causal discovery (ICML 2024).

Annadani et al,  Amortized Active Causal Induction with Deep Reinforcement Learning (NeurIPS 2024).

**Strengths And Weaknesses:**

***Strengths***

- The overall execution of the proposed framework is technically solid. The paper carefully defines the expert feedback likelihood model and incorporates posterior particle resampling through monitoring the effective sample size and rejuvenation, demonstrating methodological thoroughness.

- Grounding the method in Expected Information Gain (EIG) makes the entire feedback-driven structure learning process explicitly probabilistic. This principled formulation is appealing and conceptually well-aligned with Bayesian causal discovery.

- The theoretical result on identifiability (Theorem 7.1) under non-adversarial feedback is a welcome and meaningful contribution.

- I particularly appreciate the evaluation on CausalBench, which, to my knowledge, represents a more recent and realistic benchmark setting. Demonstrating performance in such a challenging environment strengthens the empirical validation.

***Weaknesses***

- While the conceptual execution is strong, a central practical question concerns applicability in real-world settings where the true causal graph is unknown. In many such scenarios, even expert feedback may not be substantially more informative than random guessing. The paper would benefit from a deeper discussion of how the approach performs under weak or noisy expertise.

- The related work section could be strengthened. Foundational contributions to Bayesian experimental design and EIG, such as Lindley (1956) and Chaloner & Verdinelli (1995), deserve citation. Additionally, while not identical in scope, sequential intervention design under EIG for causal discovery (e.g., Tigas et al., 2023) is relevant, especially given the particle-based methodology employed here.

- It is unclear how one should select the expert hyperparameters in practice. In the experiments, this value appears fixed at 10, which likely corresponds to a highly informative expert. Under such conditions, one might expect a relatively small number of informative queries (e.g., at most 10) to suffice. However, the reported numbers, 190 queries in the synthetic setting and 40 in other settings, seem high for a probabilistic approach with a highly informative expert. Further clarification would be helpful.

---

> ### Author Rebuttal · Authors · 2026-03-30
>
> We thank the reviewer for the positive assessment and for highlighting the technical soundness and principled use of EIG. We address the main questions below.
>
> **1. Expert reliability and practical applicability**
>
> We agree that real-world experts may be noisy or uncertain, and in some cases only weakly informative. CaPE explicitly models this via a probabilistic response model and maintains uncertainty over graph structure throughout. To assess robustness, we additionally ran a misspecified, heterogeneous expert setting where oracle responses are generated with pair-dependent reliability $(\beta^{\star}\_{\text{edge},ij}, \beta^{\star}\_{\text{dir},ij} \sim \mathrm{Uniform}[2,6])$, while inference assumes $(10,10)$.
>
> The results show that CaPE remains robust under this mismatch. In terms of ETCP (mean ± std over 10 seeds):
>
> | Policy | ETCP@1 | ETCP@100 | ETCP@150 |
> |---|---|---|---|
> | CaPE | 0.926 ± 0.002 | **0.991 ± 0.011** | **0.999 ± 0.002** |
> | UNC | 0.926 ± 0.002 | 0.986 ± 0.009 | 0.988 ± 0.008 |
> | RND | 0.925 ± 0.002 | 0.951 ± 0.004 | 0.959 ± 0.007 |
>
> Despite model mismatch, CaPE consistently outperforms uncertainty-based and random querying and achieves near-perfect posterior concentration. This indicates that the method remains effective even when expert reliability is heterogeneous or reduced.
>
> **2. Expert hyperparameters and number of queries**
>
> The choice $(\beta_{\text{edge}}, \beta_{\text{dir}})=(10,10)$ corresponds to a relatively informative but still stochastic expert (see Table 1–2). Each query provides **local information** about a single edge, while the posterior is defined over a combinatorial DAG space. Consequently, multiple queries are required to resolve global structural uncertainty, even with informative experts. We will clarify this and discuss sensitivity to hyperparameters in the revision.
>
> **3. Repeated queries and expert uncertainty**
>
> Repeated queries are naturally discouraged by the EIG objective: once a pair becomes certain under the posterior, its expected information gain decreases. Thus, CaPE focuses on unresolved edges rather than repeatedly querying the same pair. If repeated queries occur, they can be interpreted as aggregating independent judgments (e.g., from multiple experts) or repeated elicitation, both of which are common in practice. Even uncertain responses remain informative through Bayesian updating.
>
> **4. Relation to interventions and BOED**
>
> We agree that combining expert feedback with intervention design is a promising direction. CaPE focuses on a complementary regime where information is obtained through expert judgments rather than interventional data. Integrating both sources of information is a natural extension, and we will clarify this connection and discuss links to amortized intervention design approaches.
>
> **5. Evaluation metrics and related work**
>
> We agree that posterior-aware evaluation is important. In addition to SHD/AUPRC, we report entropy and ETCP, which directly capture posterior concentration and predictive quality. We will clarify this motivation and discuss alternative posterior-sensitive metrics.
>
> We will also strengthen the related work section to include foundational references on Bayesian experimental design (e.g., Lindley, 1956; Chaloner & Verdinelli, 1995), and clarify connections to recent EIG-based causal design work.
>
> **6. Limitations**
>
> We agree that explicitly discussing limitations would improve the paper. We will add a dedicated section covering assumptions on expert modeling, scalability, and practical deployment considerations.

---

> > ### Author Rebuttal · Reviewer_HoX7 · 2026-04-01
> >
> > I thank the authors for their rebuttal. Most of my concerns have been addressed. I still find the number of queries that is considered in the experiments to be high, relative to the likelihood model. I would also ask the converse: if we have a limitation of only 10 queries (say), can we a priori know what level of informativeness we would need from an expert (or the corresponding hyperparameter $\beta$ for that)?

---

> > > ### Author Response · Authors · 2026-04-04
> > >
> > > We thank the reviewer for this thoughtful follow-up.
> > >
> > > **On query budget relative to the likelihood model**.
> > >
> > > The empirical results already show substantial gains at small budgets -- for example, Table 3 (in the Appendix) shows meaningful SHD and orientation F1 improvements at T=5 and T=10 on Sachs. The full trajectories in the figures are intended to characterise the method's behaviour across a range of budgets, not to imply that large numbers of queries are typical or required.
> > >
> > > **On the inverse problem: required expert informativeness given a fixed budget**.
> > >
> > > This is an interesting question. Our theoretical analysis in Appendix D already establishes finite-time posterior concentration at an exponential rate, with the rate depending explicitly on expert quality (through the KL margin, which is directly linked to $\beta$). In principle, this relationship can be inverted to give an a priori lower bound on the expert informativeness needed to achieve a target accuracy within a fixed budget. We agree this would be a practically useful result, and we see adapting the theory more directly to answer this question as a natural and important direction for future work.
> > >
> > > A tighter, adaptive approach would be to perform a rollout of fantasised expert responses (analogous to the kriging believer strategy in Bayesian optimisation), estimating multi-step EIG under the current posterior before committing to a query sequence. For very small budgets ($T \leq 10$), this is likely tractable within our particle framework. We view this as a natural extension and will note it explicitly in the paper.

---

### Official Review · Reviewer_Vajc · 2026-03-11

**Soundness:** 2
**Presentation:** 3
**Significance:** 2
**Originality:** 2
**Overall Recommendation:** 4
**Confidence:** 3

**Summary:**

In this paper, the authors propose CaPE, a Bayesian active learning framework for expert-in-the-loop causal discovery. Given an arbitrary observational posterior over DAGs, CaPE sequentially queries an expert about pairwise edge relations, modeling their responses as three-way categorical judgments (i.e., i to j, j to i, or no edge) via a hierarchical logistic likelihood. The posterior is maintained through a particle-based approximation with importance reweighting, ESS-based resampling, and MH rejuvenation. CaPE picks which pair to ask about next by maximizing expected information gain (BALD-style) over the three response categories. On synthetic graphs, the Sachs signaling network, and CausalBench, this EIG-based query strategy recovers the underlying causal graph faster than random and uncertainty-based baselines.

**Compliance With Llm Reviewing Policy:**

Affirmed.

**Final Justification:**

This paper proposes a technically sound and well-motivated Bayesian active learning framework for expert-in-the-loop causal discovery. The authors’ rebuttal addressed my main concerns, and I raise my score from 3 to 4.

**Key Questions For Authors:**

1. Please see my comments on the Weaknesses part.

2. For the synthetic experiments, the authors construct the initial particle set by independently perturbing the ground-truth DAG. Why not use a data-driven posterior from the synthetic observational data instead? Independent perturbations may not capture the structured uncertainty (e.g., Markov equivalence) that arises from finite data.

3. How sensitive are the results to the expert hyperparameters? What happens when the expert is substantially noisier or when the assumed parameters differ from the true ones (model misspecification)?

4. Can you provide at least a qualitative or small-scale empirical comparison with the Active GFlowNet approach of da Silva et al. (2025)? Both methods iteratively refine a posterior over DAGs using expert feedback. Understanding where CaPE's particle-based SMC approach wins or loses relative to GFlowNet-based amortized inference would clarify the contribution.

5. Have you experimented with $D > 50$? What is the largest number of nodes that the proposed framework can scale to?

**Limitations:**

Yes, but only very briefly.

**Strengths And Weaknesses:**

__Strengths__:

1. This paper tries to solve an important and practical problem in scientific areas, especially relevant to biology and medicine, where interventions are expensive. I like the way that the authors target a meaningful middle ground between purely data-driven causal discovery and hard-coded prior knowledge.

2. Treating causal structure learning with expert elicitation as sequential Bayesian active learning with a probabilistic noise model is well-motivated. The proposed framework is technically coherent, and the main components: the Bayesian update, BALD-style EIG derivation, and particle-based inference (resampling + MH rejuvenation), are all reasonable and correct.

3. The paper is clearly written and well-organized, making it easy to follow.

__Weaknesses__:

1. My main concern is the lack of robustness analysis for expert model misspecification. Across all the experiments in the paper, the (semi) synthetic oracle uses the same logistic likelihood function and the set of hyperparameters. The method is never evaluated under conditions where the expert deviates from the assumed model. In practice, expert reliability would be unknown, variable across pairs, and potentially subject to systematic biases. I understand that fully addressing model misspecification may be beyond the scope of this work, but at minimum the paper should include sensitivity experiments and a more thorough discussion of this limitation.

2. The core idea of using BALD-style EIG as an acquisition function for sequentially refining causal graph posteriors is closely related to the causal Bayesian experimental design literature. [1] also employ BALD-style EIG as the acquisition for intervention target selection. I think the authors should discuss the difference between the proposed framework and this line of work, conduct benchmarking experiments if feasible, and better position the paper.

3. The paper lacks comparisons with existing expert-in-the-loop methods cited in the related work, notably Ankan & Textor (2025) and da Silva et al. (2025). The authors claim methodological advantages over these works but provide no empirical validation. Even if these methods address slightly different settings (e.g., ancestral graphs, different expert input formats), a direct comparison on a shared benchmark or a clear discussion of why comparison is infeasible would substantially strengthen the paper.

4. The identifiability theorem is mathematically reasonable but relies on strong assumptions: non-adversarial experts and every ambiguous pair being queried infinitely often. That is useful as a sanity result, but it does not strongly justify the finite-budget behavior of the actual acquisition strategy.

[1] Tigas, P., Annadani, Y., Jesson, A., Schölkopf, B., Gal, Y., & Bauer, S. (2022). Interventions, where and how? experimental design for causal models at scale. Advances in neural information processing systems, 35, 24130-24143.

---

> ### Author Rebuttal · Authors · 2026-03-30
>
> We thank the reviewer for the thoughtful and constructive feedback. Below we address the main concerns.
>
> **1. Robustness to expert model misspecification**
>
> We agree that robustness to expert misspecification is important. Our original experiments used a matched oracle to isolate the effect of sequential Bayesian updating and EIG-based querying. Importantly, CaPE is not tied to a specific expert model: inference only requires a likelihood term $p(y_{ij}\mid W)$, and both the SMC update and acquisition function remain unchanged under alternative or more expressive response models (e.g., heterogeneous or pair-dependent reliability).
>
> However, in response to the reviewer's concern, we have conducted an additional synthetic misspecification experiment where inference still assumes $(\beta_{\text{edge}}, \beta_{\text{dir}})=(10,10)$, while oracle responses are generated by a **weaker, pair-heterogeneous expert** with
> $\beta^{\star}\_{\text{edge},ij}, \beta^{\star}\_{\text{dir},ij} \sim \mathrm{Uniform}[2,6]$, testing both parameter mismatch and variability across pairs.
>
> Results show CaPE remains robust. ETCP (mean ± std over 10 seeds):
>
> | Policy | ETCP@1 | ETCP@100 | ETCP@150 |
> |---|---|---|---|
> | CaPE | 0.926 ± 0.002 | **0.991 ± 0.011** | **0.999 ± 0.002** |
> | UNC | 0.926 ± 0.002 | 0.986 ± 0.009 | 0.988 ± 0.008 |
> | RND | 0.925 ± 0.002 | 0.951 ± 0.004 | 0.959 ± 0.007 |
>
> Despite mismatch, CaPE consistently outperforms UNC and RND and achieves near-perfect posterior concentration. SHD and predictive entropy show the same trend, indicating gains are not confined to the matched setting.
>
> **2. Relation to Bayesian experimental design and intervention-based EIG**
>
> We agree EIG is central in Bayesian experimental design (BED) and will add the suggested references. However, CaPE addresses a different setting. In BOED (e.g., Tigas et al.), actions are **interventions** $a$ and observations follow $y \sim p(y \mid \mathrm{do}(a), W)$, optimizing
> $
> \mathbb{E}\_{y \sim p(y \mid \mathrm{do}(a))}\left[\mathrm{KL}(p(W \mid y, a) \parallel p(W))\right].
> $
>
> In contrast, CaPE selects **structural queries** and observes noisy expert responses
> $
> y_{ij} \sim p(y_{ij} \mid W),
> $
> with objective
> $
> \mathbb{E}\_{y\_{ij}}\left[\mathrm{KL}(q\_{t+1}(W) \parallel q\_t(W))\right].
> $
> Thus CaPE operates through a stochastic expert channel and targets structural ambiguities unresolved by data, making it complementary to BOED when interventions are costly.
>
> **3. Comparison with existing expert-in-the-loop methods**
>
> We agree such comparisons are valuable. However, prior methods operate under different formulations: Ankan & Textor (2025) use structural constraints (e.g., ancestral relations), while CaPE uses **local edge queries with a stochastic three-way model**. da Silva et al. (2025) use GFlowNets over *ancestral graphs* for *linear SEMs*, learning amortized samplers rather than sequential posterior updates. Aligning these settings would require non-trivial changes (e.g., feedback formats, inference backends). We will clarify these distinctions more explicitly.
>
> **4. Role of the identifiability result**
>
> We agree the result relies on strong assumptions and does not characterize finite-budget performance. Its role is to provide a **sanity guarantee** that the query mechanism is sufficient for identification under ideal conditions. In practice, performance is driven by the EIG criterion
> $$
> \mathrm{EIG}(i,j) = H(Y\_{ij}) - \mathbb{E}\_{W}[H(Y\_{ij}\mid W)],
> $$
> which targets epistemic uncertainty. Empirically, this yields consistent improvements over baselines. We will clarify this distinction.
>
> **5. Synthetic posterior initialization**
>
> We agree that realistic posteriors exhibit structured uncertainty (e.g., Markov equivalence). We used perturbations to **control initial uncertainty** and isolate acquisition effects. Our real-data experiments (Sachs, CausalBench) already use data-driven posteriors and show consistent gains, indicating results are not specific to this initialization. We will clarify this limitation.
>
> **Additional clarifications**
>
> Regarding scalability, our current experiments focus on $D \leq 50$, where particle-based inference remains tractable and allows accurate posterior approximation. The main computational cost scales as $O(S D^2)$ per round due to pairwise query evaluation and particle updates. While we have not yet evaluated $D > 50$ in this work, the framework is compatible with standard scaling strategies (e.g., candidate edge screening, parallelization, or amortized proposals), which we will clarify in the revision.

---

> > ### Author Rebuttal · Reviewer_Vajc · 2026-04-02
> >
> > I thank the authors for the thoughtful rebuttal and the new misspecification experiment, which partially addressed my concerns.
> >
> > However, the misspecification test varies $\beta$ within the same logistic functional form on the smallest setting (I assume that $D=20$). It demonstrates robustness to mild parameter mismatch but does not test qualitatively different expert behavior, which is closer to what real deployment would face. Full SHD and entropy results are claimed but not shown.
> > Additionally, while I acknowledge the authors' point that da Silva et al. (2025) operate on ancestral graphs rather than DAGs, both methods share the high-level goal of iteratively refining causal graph posteriors with expert feedback. A discussion of practical tradeoffs, even without direct empirical comparison, would better position the contribution.
> >
> > I would not object if other reviewers champion this paper, but given the above concerns, I maintain my original score.

---

> > > ### Author Response · Authors · 2026-04-04
> > >
> > > We thank the reviewer for the additional comments. We aim to fully address the concerns with our responses as follows.
> > >
> > >
> > > **More comprehensive misspecification experiments**
> > >
> > > To address the concern, we expanded the misspecification suite beyond mild within-family shifts. In addition to substantially noisier experts, we now include regimes that directly target qualitatively different expert behavior: `pairwise_hetero`, where reliability varies across queried pairs; `systematic_direction_bias`, where directional answers are flipped with probability `0.15`; `abstention_bias`, where directional answers are converted to `none` with probability `0.20`; and two noisy regimes, where $\beta_\text{edge}=\{1.0, 0.5\}$ and $\beta_\text{dir}=\{1.0, 0.5\}$. These settings are closer to the reviewer's concern because they model pair-dependent reliability and structured response bias rather than a single globally mis-tuned temperature. As it is challenging to show how the evaluation metrics vary with query rounds in tables, we show the additional misspecification results in figures similarly to Figure 1 of the main paper, in [this anonymous link](https://anonymous.4open.science/r/icml26_rebuttal-BE4D/ICML26_CaPE_rebuttal.pdf). We also included the full `Entropy` and `SHD` trajectories for all misspecification regimes, alongside `ETCP`.
> > >
> > > From the results, we can see that CaPE obtains better or comparable results compared with the baselines in various misspecification regimes, showing that it remains competitive under several realistic non-adversarial deviations from the assumed oracle model.
> > >
> > > We will include our response and additional results in the revision.
> > >
> > > **da Silva et al. (2025)**
> > >
> > > We agree that our method and da Silva et al. (2025) address the related broad problem of interactive/expert-in-the-loop causal discovery, and we will make this relationship more explicit in the revision. The key distinction is the representation, feedback model, and acquisition/inference design.
> > >
> > > CaPE focuses on DAGs, starts from samples of any initial posterior over DAGs, and uses an explicit three-way probabilistic expert model over local edge existence/direction together with particle-based posterior refinement and a BALD-style expected information gain acquisition rule. In contrast, AGFN operates on ancestral graphs, which allows it to represent latent confounding via bidirected edges; its expert feedback is a four-category noisy relation variable over graph relations, and its posterior refinement is built on a GFlowNet-based sampler over ancestral graphs.
> > >
> > > A second important difference is the acquisition mechanism. Our method explicitly optimizes a BALD/mutual-information-style expected information gain objective over the expert’s categorical response. AGFN discusses mutual information and information gain as natural alternatives, but ultimately uses negative expected cross-entropy as its acquisition function due to computational considerations.
> > >
> > > We will revise the paper to clarify that the practical tradeoff is therefore between a more general ancestral-graph setting with latent-confounding support on the one hand, and a modular DAG-posterior refinement framework with an explicit noisy local expert likelihood and tractable BALD-style acquisition on the other.
> > >
> > > We agree that empirical comparisons would be valuable. We attempted to reproduce da Silva et al. (2025) but encountered practical limitations: the public repository for Active GFlowNet (da Silva et al., 2025) is currently unavailable (link returns 404). We will include the comparison with the method once its code is released.

---

### Decision · Program_Chairs · 2026-04-30

**Decision:**

Accept (regular)

**Comment:**

This paper proposes CaPE, a Bayesian active learning framework for expert-in-the-loop causal discovery that refines an observational posterior over DAGs via sequential expert queries, using a three-way probabilistic likelihood and a BALD-style EIG acquisition criterion. All reviewers agreed that the proposed method is well motivated and that the work is well supported by thoery. Additionally, experimental results show strong performance with respect to baselines including at small query budgets. The authors addressed a number of concerns during the discussion period, with the majority of reviewers viewing their concerns as sufficiently addressed. The main limitations of the work as it currently stands are  scalability above 20 nodes, practical hyperparameter selection, and comparison with da Silva et al. (2025) which due to an unavailable code repository was not performed.